# Future Snow? A Spatial-Probabilistic Assessment of the Extraordinarily Low Snowpacks of 2014 and 2015 in the Oregon Cascades

Eric A. Sproles[1,2], Travis R. Roth[2], Anne W. Nolin[2]

[1]Centro de Estudios Avanzados en Zonas Áridas, Universidad de La Serena, Raul Bitran 1305, La Serena, Chile
[2]College of College of Earth, Ocean, and Atmospheric Sciences; Oregon State University, 104 CEOAS Admin Bldg, Corvallis, OR, 97331-5503, USA

*Correspondence to*: Eric A. Sproles (eric.sproles@gmail.com)

**Abstract.** In the Pacific Northwest, USA, the extraordinarily low snowpacks of winters 2013–2014 and 2014–2015 stressed
regional water resources and the social-environmental system. We introduce two new approaches to better understand how seasonal snow water storage during these two winters would compare to snow water storage under warmer climate conditions. The first approach calculates a spatial-probabilistic metric representing the likelihood that snow water storage of 2013–2014 and 2014–2015 would occur under +2°C perturbed climate conditions.  We computed snow water storage (basin-wide and across elevations), and the ratio of snow water equivalent to cumulative precipitation (across elevations) for the
McKenzie River basin (3 041 km$^2$), a major tributary to the Willamette River in Oregon, USA. We applied these computations to calculate the occurrence probability for similarly low snow water storage under climate warming. Results suggest that, relative to +2°C conditions, basin-wide snow water storage during winter 2013–2014 would be above average while that of winter 2014–2015 would be far below average. April 1 snow water storage corresponds to a 42% (2013–2014) and 92% (2014–2015) probability of being met or exceeded in any given year. The second approach introduces the concept
of snow analogs to improve the anticipatory capacity of climate change impacts on snow derived water resources. The use of a spatial-probabilistic approach and snow analogs provide new methods of assessing basin-wide snow water storage in a non-stationary climate, and are readily applicable in other snow dominated watersheds.

## 1 Introduction

In the Pacific Northwest (PNW), mountain snowpacks during the winters of 2013–2014 and 2014–2015 were at or near
record lows and well below 50% of the historic median value (National Resource Conservation Service, 2014, 2015b). For several decades the Natural Resources Conservation Service (NRCS) Snowpack Telemetry (SNOTEL) network has provided measurements of snow water equivalent (SWE; the amount of water contained within the snowpack) and meteorological data. These station-based measurements have historically served as a proxy for basin-wide snow storage and provide an effective SWE index for estimating streamflow, however under a shifting climate these statistical relationships have also

changed (Montoya et al., 2014). The PNW's extreme low snowpacks and subsequent snow water storage of 2013–2014 and 2014–2015 highlight the limitations of location-specific measurements in a shifting climate.

On March 1, 2015, 47% of snow monitoring sites in the Willamette River Basin (WRB, 29 730 km$^2$, Fig. 1) registered zero SWE while snow was still present at higher elevations. The absence of snow during the winter of 2014–2015 stands in
contrast to cumulative winter precipitation, which was at 83% of normal (778 mm) for November to February (derived from PRISM data (Daly et al., 2008). While the concurrent drought in California received substantial attention, the economic and environmental impacts in the PNW were also profound. These two extreme low snowpacks in the PNW led to ski area closures, recreation restrictions, municipal water limitations, severe wildfires, low streamflows, nearly dry reservoirs, harmful algal blooms, and high fish mortality (Associated Press, 2015; Bend Bulletin, 2015; Oregon Department of Fish and
Wildlife, 2015; The Oregonian, 2015a; The Oregonian, 2015b). These types of externalities highlight the importance of mountain snow water storage and the implications of snow drought.

Mountain snow water storage in the western Oregon Cascades and across the western United States serves as vital inter-seasonal storage from cool, wet winters with low water demand to hot, dry summers when demand peaks (Oregon Water Supply and Conservation Initiative, 2008; United States Army Corps of Engineers, 2001). The western Oregon Cascades
form the eastern boundary of the WRB (Fig.1), and abundant winter precipitation falling in these mountains (up to 3000 mm yr$^{-1}$) sustains the 13$^{th}$ highest streamflow in the conterminous United States (Hulse et al., 2002). Even in such a wet place, snowmelt is critically important. Brooks et al. (2012) estimated that over 60–80% of summer base flow in the Willamette River derives from the snow zone at elevations over 1200 m, though this elevational zone represents only 12% of the land area and 15.6% of the annual precipitation in the basin.

The McKenzie River Basin (MRB, 3 041 km$^2$) is a major tributary to the WRB (Fig. 1), and is located in the main part of the Willamette's "at-risk" snow zone (Nolin and Daly, 2006). Snowmelt in MRB is critical to meeting environmental and societal demands of the WRB, supplying almost 25% of the river's summer discharge at its confluence with the Columbia River near Portland, Oregon (Hulse et al., 2002), despite only occupying 10% of its area. The hypsometry of the MRB and WRB are visually similar (Fig 1b) and statistically similar when tested using a two-parameter Kolmogorov-Smirnov test for
sample distribution (Young, 1977).

The maritime snowpacks of the MRB, WRB, and the PNW are deep (>1.5 m), relatively warm (Sturm et al., 1995), and SWE typically reaches its basin-wide maximum on approximately April 1 (Serreze et al., 1999; Stewart et al., 2004). Nolin and Daly (2006) identified snow in the WRB as climatologically "at-risk" since it typically accumulates at close to 0ºC and can convert to rainfall with just a slight increase in temperature. As a result of changes in circulation patterns and warmer
temperatures there have been declines in April 1 SWE in the PNW (Barnett et al., 2005; Kapnick and Hall, 2012; Luce and Holden, 2009; Mote, 2006; Mote et al., 2005; Service, 2004; Stoelinga et al., 2010), and peak streamflow has shifted to earlier in the year (Fritze et al., 2011; Stewart, 2009).

These shifts in streamflow highlight the challenges of using location specific measurements of SWE for prediction in changing climate. While SNOTEL sites provide valuable and robust data, they typically occupy a limited elevation range

that leads to an under-sampling of both the high elevation snow zone and the lower elevation rain-snow transition zone (Molotch and Bales, 2006; Montoya et al., 2014; Nolin, 2012). This limited range holds true in the MRB, where the mean elevation is 1424 m and the elevational range between the five stations is only 245 m.

Elevational shifts in snowpack accumulation due to observed temperature increases make the past less representative of the future (Dozier, 2011; Milly et al., 2008). Additionally, patterns of snow accumulation and melt in the PNW vary as non-linear functions of elevation, slope, aspect, and landcover (Tennant et al., 2015). Augmenting point-based measurements of SWE with metrics that effectively estimate snow water storage in a mountain landscape would include calculations for normal and extreme years across elevations and at the basin scale—especially under current climate trends (Dozier, 2011).

The dimensionless ratio of SWE to precipitation (SWE:P) represents the proportion of snow water equivalent relative to cumulative precipitation (snowfall plus rainfall) over a specified time interval (Serreze et al., 1999). This ratio normalizes snow water storage by cumulative precipitation, emphasizing the impacts of temperature on snowpack accumulation and melt. When computed for April 1, the time of year when maximum basin-scale SWE is considered to occur, this ratio can be an effective measure of the stages of accumulation and melt (Clow, 2010).

Understanding how relationships between snowpack, precipitation, and temperature will be expressed at the basin scale is particularly important in the maritime PNW. Physically-based modelling studies of climate impacts in the PNW describe reduced snow water storage and earlier streamflow across the region (Elsner et al., 2010; Hamlet, 2011; Sproles et al., 2013). These deterministic approaches provide a range of outputs of past and future conditions. However these approaches stop short of an analog approach that links an individual year from the past, particularly a low snow year, to projected conditions. Climate analogs serve as a useful device to examine potential impacts on societally relevant of predictands (e.g. forest health, environmental flows, municipal water supply), and applies previous conditions to represent potential future conditions (Hallegatte et al., 2007; McLeman and Hunter, 2010; Ramírez-Villegas et al., 2011; Webb et al., 2013). For example, Ramírez-Villegas et al. (2011) developed analogs of climate and agricultural practices to identify prior climatic events that may provide insights into the impacts of future climate change in both time and space.

Incorporating an analog approach allows planners and managers to develop anticipatory capacity, the ability to better anticipate changing scenarios as needs and context change over time (Nelson et al., 2008; Rhodes and Ross, 2009). Using the extreme low snow water storage of 2014-2015 as an example, residents of the Willamette Valley raised concerns regarding the safety and taste of domestic drinking water during the summer months. These changes in water characteristics led public works departments to examine future strategies and equipment to mitigate future water quality concerns (Hall, 2015). From a hydrological perspective, this same analog approach is also used in describing streamflow, and is most commonly framed using statistical metrics. For example, the spatial extent for a previous 100-year flood event serves as an analog of floodplain dynamics and provides anticipatory capacity for land use planners and water managers.

Based on the premise that future snow water storage conditions will resemble previous winters that were warm, Luce et al. (2014) developed spatial and temporal analogs of snow water storage sensitivity to temperature and precipitation across the western United States using point-based SNOTEL data. Similarly, Cooper et al. (2016) applied model-based analyses to

compare the winters of 2014 and 2015 to projected future conditions using individual metrics of snowpack (Snow Disappearance Date, Date of Peak SWE, and Duration of Snow Cover) at SNOTEL locations in the Oregon Cascades. This approach is informative, even though point-based analysis in projected warmer conditions may not represent basin-wide conditions (Dozier, 2011; Milly et al., 2008), specifically as the rain-snow transition shifts towards higher elevations (Nolin et al., 2012; Nolin and Daly, 2006).

To develop statistically valid analogs for snow water storage and snow water storage at the basin scale requires a spatially explicit, probabilistic approach that calculate the statistical likelihood of SWE across a topographically complex mountain basin. For example, to address the question "What is the likelihood that the snow droughts of WYs 2014 and 2015 will occur in the future?" can be addressed by developing statistical thresholds of SWE and SWE:P with regards to time and location. This spatial-probabilistic approach develops upper or lower limits of predicted snow water storage conditions throughout a watershed. While probabilistic approaches are common to streamflow hydrology, spatial approaches to probabilistic questions are less common. A notable application of a spatial-based, probabilistic approach was developed by Graf (1984). This research applied 107 years of channel migration records to calculate the probability of subsequent erosion in a given parcel, creating a probabilistic map of river movement. The map outlined the character of the river system that identified areas where channel migration was more likely to occur. Margulis et al. (2016) characterized the extreme California snow deficit of 2015, but did not compare this snow drought to potentially warmer climatic conditions. Snow hydrology models can readily incorporate climate change projections (Adam et al., 2009; Sproles et al., 2013) and model outputs can be assessed using a spatial-probabilistic framework that explicitly accounts for elevation.

This research introduces a physically based, spatial-probabilistic modelling framework to compare the extraordinarily low snow winters of WY 2014 and WY 2015 (WY=Water Year, defined as 1 October – 30 September in the western United States) in the context of warmer climatic conditions. Our approach captures the spatial variability of mountain snow water storage under warmer temperatures across decades by simulating the variability of SWE and SWE:P at the basin scale for 23 WYs using +2°C conditions. These outputs are used to frame the snow water storage of WY 2014 and WY 2015 in the context of future snow and snow analogs. This approach is intended to build anticipatory capacity for climate change impacts in the PNW through snow analogs. While limited to the McKenzie River Basin (a well-studied watershed that is characteristic for maritime snow in the WRB (Nolin and Daly, 2006), regional sensitivity to climate warming makes PNW snowpack and snow water storage, and those in similar maritime climates, acutely vulnerable to snow drought (Leibowitz et al., 2014; Nolin and Daly, 2006).

Specifically, we ask:

- How does snow water storage from WY 2014 and WY 2015 compare to snow water storage under +2°C conditions?

- What is the probability that similar snowpacks and snow water storage will occur in the future?

- How does snow water storage during WY 2014 and WY 2015 vary by elevation?

## 2 Research Methods

Our approach applies a spatially-distributed and physically-based snow hydrology model to compute probabilities of SWE and SWE:P for 23 WYs under +2C winter conditions. We then model WY 2014 and WY 2015 snow water storage and these outputs provide probabilistic context for the snow water storage of those two winters. Below we provide details on the study
area and specific methods used in this approach.

This study focuses on the McKenzie River Basin. In addition to the MRB being a major tributary to the Willamette River, it has a well-developed network of meteorological stations associated with the HJ Andrews Long Term Ecological Research site, four SNOTEL stations, four dams for flood control and hydropower, serves as the primary source of domestic water for 200 000 people, and is home to federally protected salmonids, amphibians, and mussels. The MRB is characterized by wet
winters and dry summers, with average annual precipitation ranging from 1000 mm to 3000 mm that follows the elevation gradient (114–3147 m). Elevations between 1000 and 2000 m comprise 42% of the MRB's total area (Fig. 1a) and 93% of the total snow water storage in the MRB (Sproles et al., 2013). While elevations above 2000 m accumulate the most SWE per unit area, that zone comprises only 1% of total area and 6% of the total snow water storage for the MRB. In terms of volume, snow is the primary seasonal water storage mechanism in the MRB with historic mean basin-wide snow water
storage (SWE × area; 1989–2009) of 1.26 $km^3$ on April 1 (Sproles et al., 2013), compared with total reservoir storage of 0.40 $km^3$ (United States Army Corps of Engineers, 2016; United States Department of Agriculture, 2016). By comparison groundwater storage for the MRB was estimated to be roughly 4 $km^3$, with a mean transit time of seven years (Jefferson et al., 2006).

Spatially-distributed values of precipitation and SWE were computed using SnowModel (Liston and Elder, 2006a, 2006b)
for WY 1989–2012. SnowModel is a spatially distributed, process based model that computes temperature, precipitation, and the full winter season evolution of SWE including accumulation, canopy interception, wind redistribution, sublimation, evaporation, and melt. The model framework applied in this study is the same as applied in Sproles et al. (2013), with the addition of a multi-layer snowpack algorithm. Because the modelling framework is physically-based and spatially-distributed, perturbations to temperature inputs will propagate throughout the model including absolute humidity and energy
balance calculations, thus maintaining the dependencies between snowpack and temperature. WY 2005 was excluded due to prolonged regional temperature inversions that were not resolved in the model (Sproles et al., 2013).

Model input data were derived from SNOTEL and station data within the study area (six stations in total), nearly spanning the full elevation range of the MRB (Fig. 1; Sproles et al., 2013). The 23-year set of model forcing data includes winters with above average, normal, and below average snowpack; positive, negative, and neutral ENSO climate patterns; and cool and
warm phases of the Pacific Decadal Oscillation (Brown and Kipfmueller, 2012). The model was run at a daily time step and 100-m grid resolution. In the calibration and validation phase, the model was first calibrated to temperature and precipitation to ensure that the model results were representative of these first order inputs, with mean Nash-Sutcliffe Efficiencies (Legates and McCabe, 1999; Nash and Sutcliffe, 1970) of 0.80 and 0.97 respectively. The model was then calibrated for

physical snowpack conditions (mean Nash-Sutcliffe efficiency of 0.83 for automated stations, 0.70 for field locations, and an overall spatial accuracy of 82% compared with Landast fSCA data). For a detailed description of the model structure, calibration, validation, and performance please refer to Sproles et al. (2013).

Using the validated model, we increased temperatures by +2ºC and re-ran the model over the same timeframe and spatial domain. Projections for future precipitation in the WRB and the PNW are highly uncertain (Safeeq et al., 2016), and in the Oregon Cascades temperature, not precipitation, dominates the accumulation and melt cycles of snowpack (Sproles, 2012; Sproles et al., 2013). Our delta increase to temperature is intended to be straightforward, and to avoid the uncertainties associated with precipitation in this region.

We extracted SWE and precipitation (P) data, and computed 5-day averages for each centered on the first day of each month for January to June, for every year in the model run, and for each grid cell in the model domain. These 5-day mean values were used to minimize any effects from individual events (melt, snowfall) while still capturing the overall snow water storage characteristics at the beginning of the month.

Exceedance probability (EP) is a widely used hydrologic metric describing the statistical likelihood that a value of a given magnitude or greater will occur in a specified time period (e.g. annually) (Sadovský et al., 2012; Salas and Obeysekera, 2013). Expressed as a percentage, it is calculated as:

$$EP = \left(\frac{m}{n+1}\right) \times 100 \tag{1}$$

where, $m$ is the rank of the data value (ranked from highest to lowest) and $n$ is the total number of data values (Dingman, 2002).

For example, 20% EP (a low annual exceedance probability) is the statistical likelihood that a value could be met or exceeded 20% of the time, or a 1 in 5 chance of occurring or being exceeded in any year. 20% EP represents a relatively large value. A 90% EP (a high annual exceedance probability) describes the statistical likelihood of a measurement that would be met or exceeded in 90% of the time, and represents a relatively low value. EP is commonly applied to point-based data such as a stream gage or SNOTEL station.. However, because mountain snow water storage varies by elevation, slope, aspect, and landcover (Tennant et al., 2015), we expanded point-based EP calculations to the watershed scale to include normal and extreme years.

To accomplish a spatial perspective of exceedance probability, we applied 23 years of model output to compute the EP for the first of the month (January to June) based upon the 5-day averaged SWE and SWE:P values for each grid cell in the model domain. The dimensions of the model domain is a grid of 759 rows × 1121 columns. In order to sort each grid cell individually across the 23 datasets (years), the two-dimensional data sets (759 rows × 1121 columns) was decomposed into 23 one-dimensional vectors (1 × 850,839) then combined to create a 23 × 850,839 matrix. The location information of each grid cell was retained for subsequent mapping and analysis. For each year, the 23 values in each row were sorted from

highest to lowest. The $23 \times 850{,}839$ data matrix was recomposed into 23 data matrices of dimension $759 \times 1121$ creating a corresponding spatial exceedance probability matrix. This was completed for each month (January to June).

To respond to the question "How do snow water storage from WY 2014 and WY 2015 compare to snow water storage under a warmer climate?" we modeled SWE and SWE:P using SnowModel with meteorological forcing data from WY 2014 and WY 2015 for the MRB, with the same stations as from our previously validated model runs. These model runs were also validated using the same methods as described in Sproles et al. (2013). We then compared the snowpack metrics from these two winters with model output from the +2°C climate scenario.

Elevation is the most important physiographic variable in determining SWE in this basin (Nolin, 2012) so we aggregated the data into 50-m elevation bands (Fig.1a). In each of these bands we computed snow water storage (km$^3$) and mean SWE:P (m/m). This allowed us to understand the variation of snowpack properties by elevation, their spatial probability of occurrence, and the statistical context for the extraordinary snowpacks of WY 2014 and WY 2015.

*An important point to bear in mind is that the EP values were computed using perturbed meteorological forcing data (+2°C), while values for WY 2014 and WY 2015 were derived from unperturbed meteorological forcing data.*

## 3 Results

For context, historically in the MRB 62% of precipitation falls in the Nov to Mar (N–M) time period as calculated from monthly precipitation data from 30-year PRISM gridded climate normals (Daly et al., 2008). Within that period, Dec to Feb (DJF) are historically the coldest and wettest months (Daly et al., 2008). For N–M in WY 2014, precipitation was at 102% of the 30-year normal (calculated from PRISM data) and temperatures at SNOTEL stations in the MRB were 0.9ºC warmer than normal (National Resource Conservation Service, 2015a). For the DJF period, WY 2014 monthly precipitation was 96% of normal and SNOTEL temperatures were 0.7ºC warmer than normal. During WY 2015, N–M precipitation was 81% of the 30-year average but temperatures in the snow zone were 2.7ºC warmer than average. For the DJF period of WY 2015, monthly precipitation was 78% of normal and temperatures in the snow zone were 3.3ºC warmer than normal (National Resource Conservation Service, 2015a). To provide historical context, Fig. 2a and b presents graphically presents the 30-year precipitation and temperature normals from the PRISM datasets as compared to WYs 2014 and 2015. Fig. 2c presents modeled basin-wide snow water storage for WYs 2014 and 2015 as compared to the 23-year mean from the +2ºC snowpack simulations.

A warmer than normal Jan 2014 limited snowpack accumulation during the early portion of the winter, and wetter than normal conditions in Feb 2014 accompanied by near normal mean temperatures increased basin-wide snow water storage to near average/above average snowpack conditions (as compared to a +2ºC perturbation) for the remainder of the season (Fig. 2c). The warmer than normal conditions that persisted throughout WY2015 greatly inhibited seasonal snowpack accumulation, despite above average precipitation in Mar 2015 (Fig. 2c). For a more detailed long-term climate analysis please refer to Abatzoglou et al. (2014).

### 3.1 Snow Water Storage

In the context of our exceedance probability framework, we see that the April 1 basin-wide snow water storage for WY 2014 falls between the 42 and 46% EP, meaning that WY 2014 snow water storage is slightly above average for a +2ºC model perturbation (Figs. 2, 3, 4, and 5a and c). Snowfall occurring after April 1, 2014 improved late season snow water storage

corresponding to 33% and 25% EP for May and June, respectively (Figs. 3 and 4). In WY 2015 basin-wide snow water storage was well below historical conditions, even when compared with +2ºC conditions. April 1 snow water storage for WY 2015 corresponds to 92% EP (Figs. 3, 4, 5b, and 5d). In that year, there was little late spring snowfall, so unlike WY 2014 basin-wide snow water storage did not increase (Fig. 3). WY 2015 was also notable in that peak snow water storage occurred in January and was only 0.21 km$^3$, corresponding to 79% EP (Figs. 3 and 4).

Fig. 4 shows the spatial exceedance probabilities for the +2ºC model runs, aggregated into 50-m elevation increments (WY2014, 42% EP; WY2015, 92% EP). For most years, the total amount of April 1 snow water storage is greatest within the elevation range of 1300–1800 m. However in WY 2015 this mid-elevation zone (1300–1800 m), representing 393 km$^2$ (as calculated from the elevation dataset) is essentially snow-free (Fig. 4). Snow water storage in this elevation range is critical for late season runoff, as 1200 m represents the elevation threshold for summer baseflow contributions (Brooks et al.,

2012). From a spatial perspective, Fig. 5 presents the distribution of SWE in the MRB in WYs 2014 and 2015 on April 1, as compared to the 46% and 92% EP (as compared to a +2ºC perturbation), respectively. These figures show snow water storage is almost entirely limited to the upper portions of the basin, and that the more spatially extensive mid-elevations where snow accumulates historically are snow free. In other words, in WY 2014 and 2015 the zone where snowmelt has historically contributed most to groundwater recharge (Jefferson et al., 2008; Tague and Grant, 2009), shifted to rain.

Jefferson et al. (2008) showed that the recharge signal varies spatially and temporally, and that the location of the rain-snow transition is the dominant control on recharge for at the watershed scale.

### 3.2 SWE:P

This elevation dependent shift from rain to snow is evident in Fig. 6, where at an elevation of 1200 m, SWE:P is below 0.06 for the period January to June in both WY 2014 and 2015. This ratio does not exceed 0.20 until an elevation of 1500 m in

WY 2014, which is still markedly lower than the long-term mean SWE:P at the McKenzie SNOTEL site (0.58, 1454 m). In WY 2015 this 0.20 threshold is not reached until an elevation of 1750 m, approximately 300 m above the highest elevation SNOTEL site in the MRB, and thus was not captured in the SNOTEL data. From February to May in WY 2014, SWE:P increases due to late season storms that added snow water storage, and remained above 50% EP when compared with +2ºC conditions. From February to May in WY 2015, SWE:P never surpasses the 0.60 threshold, and remains below 90% EP

when compared with +2ºC conditions.

## 4 Discussion and Conclusion

The winters of 2014 and 2015 had very low snowpacks across the Pacific Northwest due to higher than normal winter temperatures but average or near-average precipitation (Fig. 2, *National Resource Conservation Service*, 2014, 2015b), highlighting the sensitivity of the region's snowpack to increased temperature. In the MRB snow zone mean temperatures (Nov-Mar) were 0.9ºC above the 30-year normal in WY 2014, while WY 2015 were 2.7ºC above normal. These low snow years persisted even under normal and slightly below normal Nov-Mar precipitation (WY2014, 102%; WY2015 81%). The SWE:P metric also identifies increased temperature, rather than reduced precipitation, as the primary reason for the diminished snow water storage of WY 2014 and WY 2015, especially at mid elevations. At 1500 m the April SWE:P values for the two years are considerably different (Fig. 6; WY2014, SWE:P = 0.22, 60% EP; WY2015, SWE:P = 0.04, 95% EP).

As such, these two winters' extraordinarily low snowpacks offer an analog perspective for projected future snow conditions in the MRB and potentially the Willamette River Basin. WY 2014 serves as a snow analog for slightly warmer conditions (+1ºC), with an EP between 42 and 46%. While WY 2015 serves as a snow analog for conditions increasing beyond 2.5ºC with an EP of around 92%. The volumetric difference between the two years is considerable (0.56 km$^3$), representing 1.4 times more than the total reservoir storage capacity of the MRB (United States Army Corps of Engineers, 2016; United States Department of Agriculture, 2016).

The SWE:P metrics across elevation bands provides a simple yet telling description of precipitation phase (rainfall vs. snowfall) and evolution of snow water storage (accumulation and ablation). The shifts from rain to snow seen in the modelled results highlight the limitations of a monitoring network that occupies a limited range. In the MRB the SNOTEL stations occupy a mean elevation of 1424 m with a range of only 245 m. During WY 2014 and 2015, this limited range did not capture zones with maximum snow water volume and were essentially below the rain-snow transition (Figs. 4 and 6). This same under representation of snowpack was found throughout the greater WRB with 47% of snow monitoring sites registering zero SWE while snow was still present at higher elevations on March 1, 2015.

As precipitation shifts from snow to rain, the SWE:P metric can augment individual values of SWE and P to provide key information on shifts in water storage throughout the course of a winter, and valuable insights to water resource managers in a non-stationary climate. For example, on March 1, when basin wide SWE is typically approaching its maximum, both years are essentially snow free at 1200 m. A low SWE:P ratio in March under normal winter precipitation conditions could indicate peak streamflow has occurred or most likely would occur earlier in the year, which has important implications for water resource management in subsequent months.

Low snow water storage and shifts in streamflow negatively impact water quantity, water quality, hydropower operations, winter snow sports, and summer recreation. In WY 2015, record low snow water storage led to summer drought declarations, extreme fire danger, and modified hydropower operations in the MRB. The typical consistent flow of the groundwater-fed McKenzie River was at 63% of August-September median flow (United States Geological Survey, 2015). Hoodoo Ski Area,

located at Santiam Pass, was open for only a few weekends in WY 2014 and in WY 2015 they suspended operations in mid-January, the shortest season in their 77-year history. In the adjacent Santiam River Basin (north of the MRB), diminished snow water storage and less-than-anticipated spring rains in WY 2015 pushed the Detroit Reservoir (storage capacity 0.35 $km^3$) to historic low levels. In May harmful blue-green algae concentrations were above acceptable amounts by seven-fold, and July reservoir levels were approximately 21-m below capacity. Concerns over the taste and safety of domestic drinking water in the Willamette Valley prompted municipal water managers to explore options for upgrading water treatment facilities.

At more broad timescales the shift from snow to rain at mid-elevations could also potentially impact groundwater recharge. The rain-snow transition is the dominant control on recharge in the MRB, and varies spatially and temporally (Jefferson et al. 2008). Because groundwater storage is large and transit times in the MRB are approximately 7 years (Jefferson et al. 2008), the full impacts of WY2014 and WY2015 on ground and surface water resources are not yet known.

Water quality, energy production, and recreation externalities are not well represented in deterministic models, but become challenging realities that the public faces in years with low snow. Intervention strategies can fail because they lack adequate information about the impacts of climate change that are not incorporated into deterministic physical models and play out at the human scale (Ramírez-Villegas et al., 2011). Transitioning from purely deterministic approaches (i.e. snow water storage is reduced by a certain percentage) to ones that link climate and snow conditions with real world impacts provide a complementary perspective for mitigation and adaptation. Our analog approach combines projected climate impacts with the extreme low snow years of 2013-2014 and 2014-2015 for insights into improved management in shifting conditions. Such an analog approach allows planners and managers to develop adaptation and mitigation strategies that use the past to demonstrate what did or did not work under climate stress, and help build a more informed understanding of ways to improve future planning efforts (Ramírez-Villegas et al., 2011).

Climate change impacts are often expressed in probabilistic terms (Randall et al., 2007) and so it is logically consistent to estimate snowpacks and snow water storage in this manner. This research does not assume that the probabilities presented here are based upon a precise representation of future conditions nor that future climates will be +2ºC warmer every winter. We present these results as a way to frame the likelihood of future basin-wide snow water storage in the context of our current understanding of climate change. These probabilistic insights are then used to identify WY 2014 and WY 2015 as analogs years for managers and decision makers. The WY 2014 snow water storage would be slightly above average for +2ºC conditions; and the WY 2015 snow water storage would be very low snow water storage for +2ºC conditions, but not a record low. These analog years thus provide guidance for adaptation strategies to mitigate potential failures of existing management plans.

Our spatially explicit approach augments information from the existing SNOTEL network. While SNOTEL data continue to play a key role for seasonal streamflow forecasting under historic climatic conditions, these statistical relationships have been changing (Montoya et al., 2014). While providing modern scientific equipment, SNOTEL sites in the MRB occupy a limited range (245 m) in the mid-elevations and may not capture basin-wide snow water storage in warmer conditions. For

example, in the MRB all SNOTEL sites in the MRB were snow-free for most of February to March 2015 and therefore incapable of providing predictive skill for water resource management. Our basin-scale probabilistic approach provides a more complete picture of water storage and captures the elevation variability absent in point-based measurements.

The winters of WY 2014 and WY 2015 demonstrate a considerable departure from the stationary snow water storage conditions on which present-day management plans are based. With continued current warmer climates, the snow water storage conditions represented by these two winters are more likely to occur. In the meantime, the value of spatially explicit probabilistic calculations rests in the ability to better define the range of statistical outcomes of subsequent winters that are representative of basin-wide conditions. Framing the low snow water storage of WY 2014 and WY 2015 as analogs of future snow provides insights into potential climate impacts and externalities on social and environmental systems. Together, probabilistic metrics and snow water storage analogs can help build capacity to better anticipate hydrologic changes in a warming climate.

**Acknowledgements**

This research was made possible by support from the National Science Foundation (BCS-0903118 and EAR-1039192). We gratefully acknowledge the modelling guidance of Dr. Glen Liston.  The data for Figs. 3, 4, and 6 can be downloaded at http://people.oregonstate.edu/~sprolese/snow_frequency/. The authors would also like to thank the two anonymous reviewers for their comments and expertise the Associate Editors of the Cryosphere for managing the submission and revision process.

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

**Figures:**

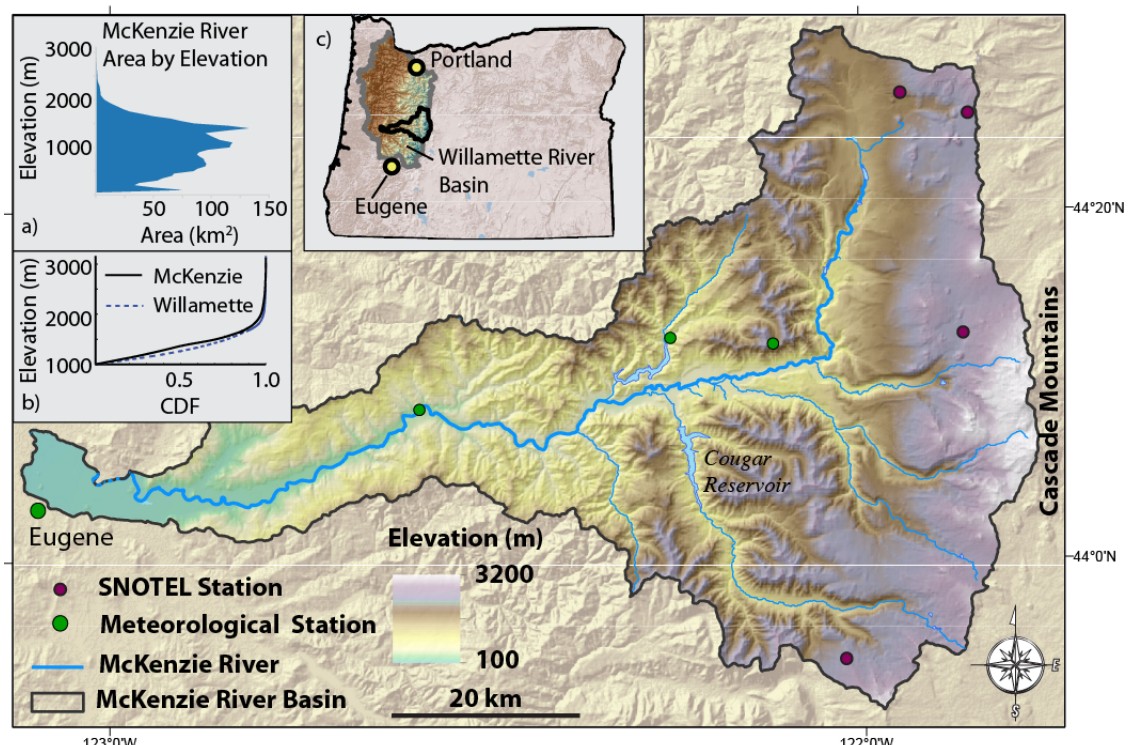

**Figure 1: Context map of the McKenzie River basin, and its geographic relationship to the Willamette River basin. The geographic locations of the SNOTEL other meteorological stations used as model forcings show the altitudinal range of inputs. Inset figure a) represents the area by elevation for the McKenzie River basin Inset figure b) presents the Cumulative Distribution Functions (CDF) for the elevation of the Willamette and McKenzie River Basins for elevations above 1000 m, and is separated into 50m bins.**

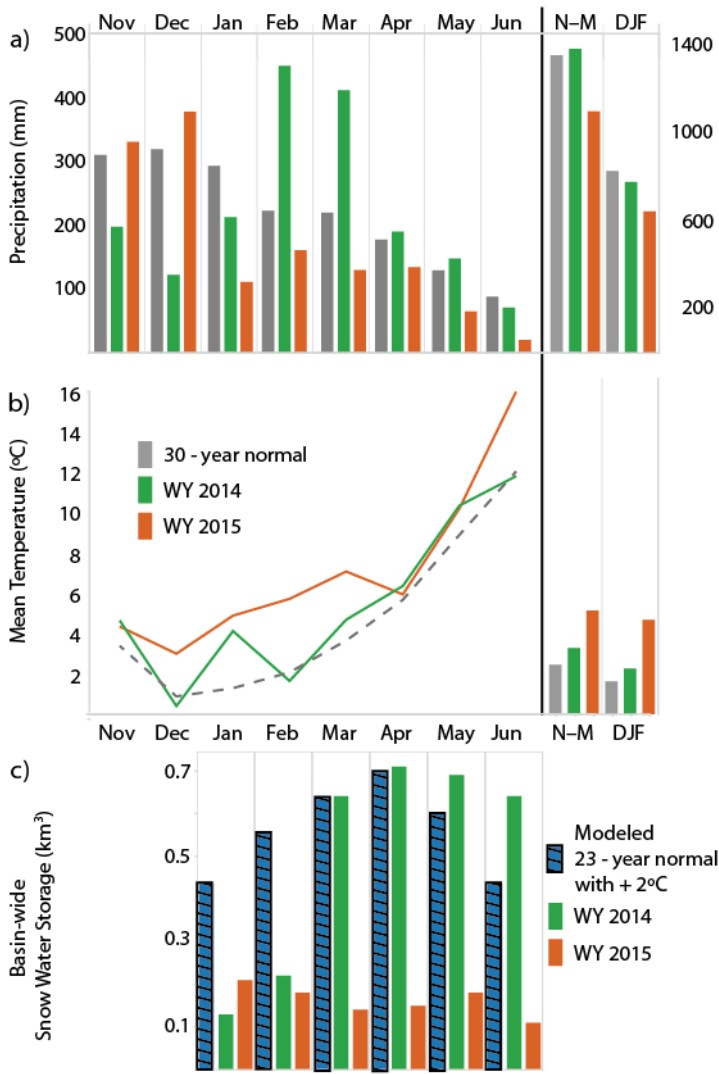

**Figure 2: The total precipitation (a) and mean temperatures (b) for the McKenzie River basin for water years 2014 and 2015 as compared to the 30-year normal (from the PRISM datasets). The lower figure (c) represents Basin-wide Snow Water Storage for the McKenzie River Basin for water years 2014 and 2015 and the normals (+2ºC) calculated from the 23 years used in this study. The calculations for snowpack are 5-day averages centered on the first day of each month.**

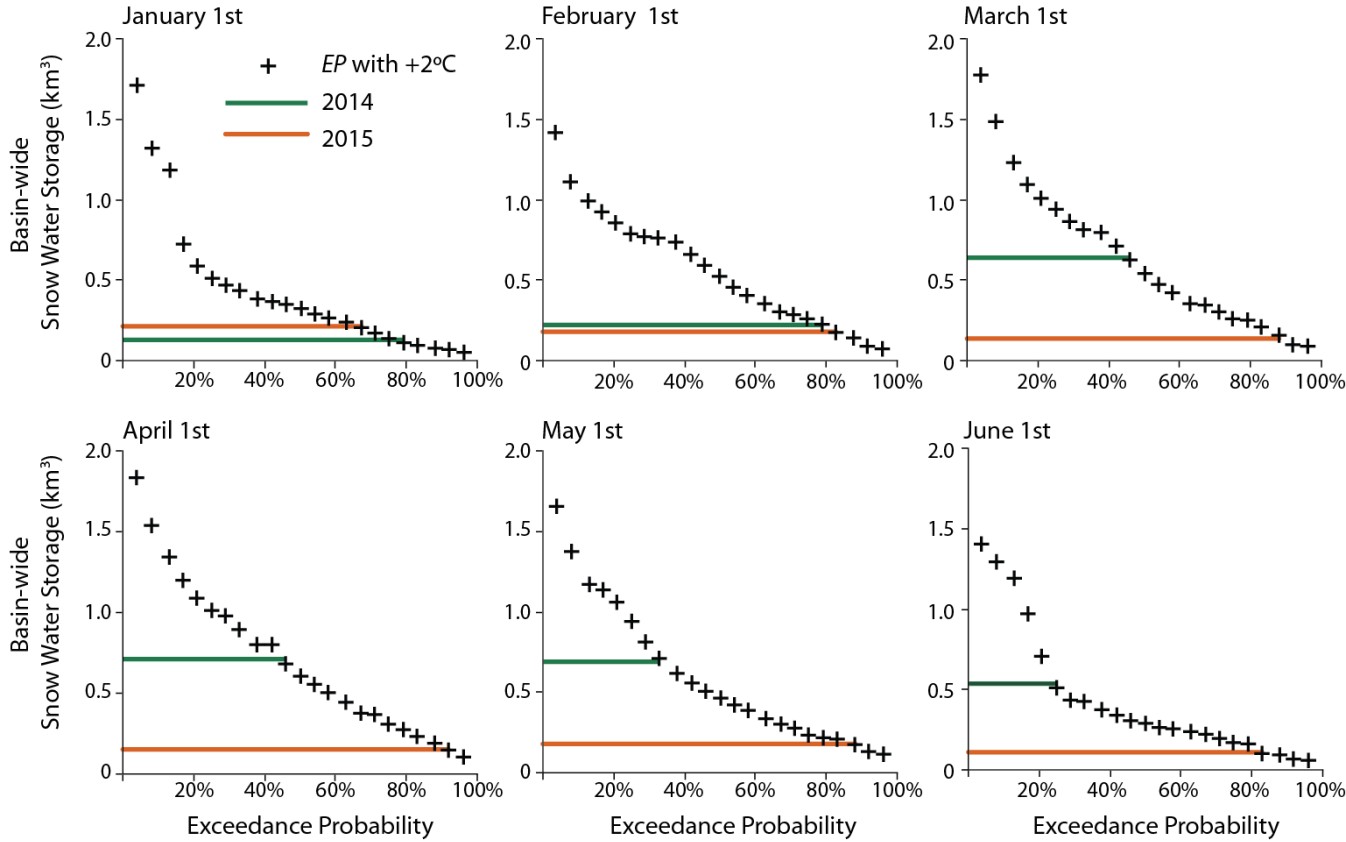

**Figure 3: The exceedance probability of basin-wide snow water storage under +2ºC conditions. During 2014 snow water storage increased considerably in March to reach above average conditions. The snowpack during the winter of 2015 was extremely low, and never increased beyond 0.21 km³. The calculations are 5-day averages centered on the first day of each month.**

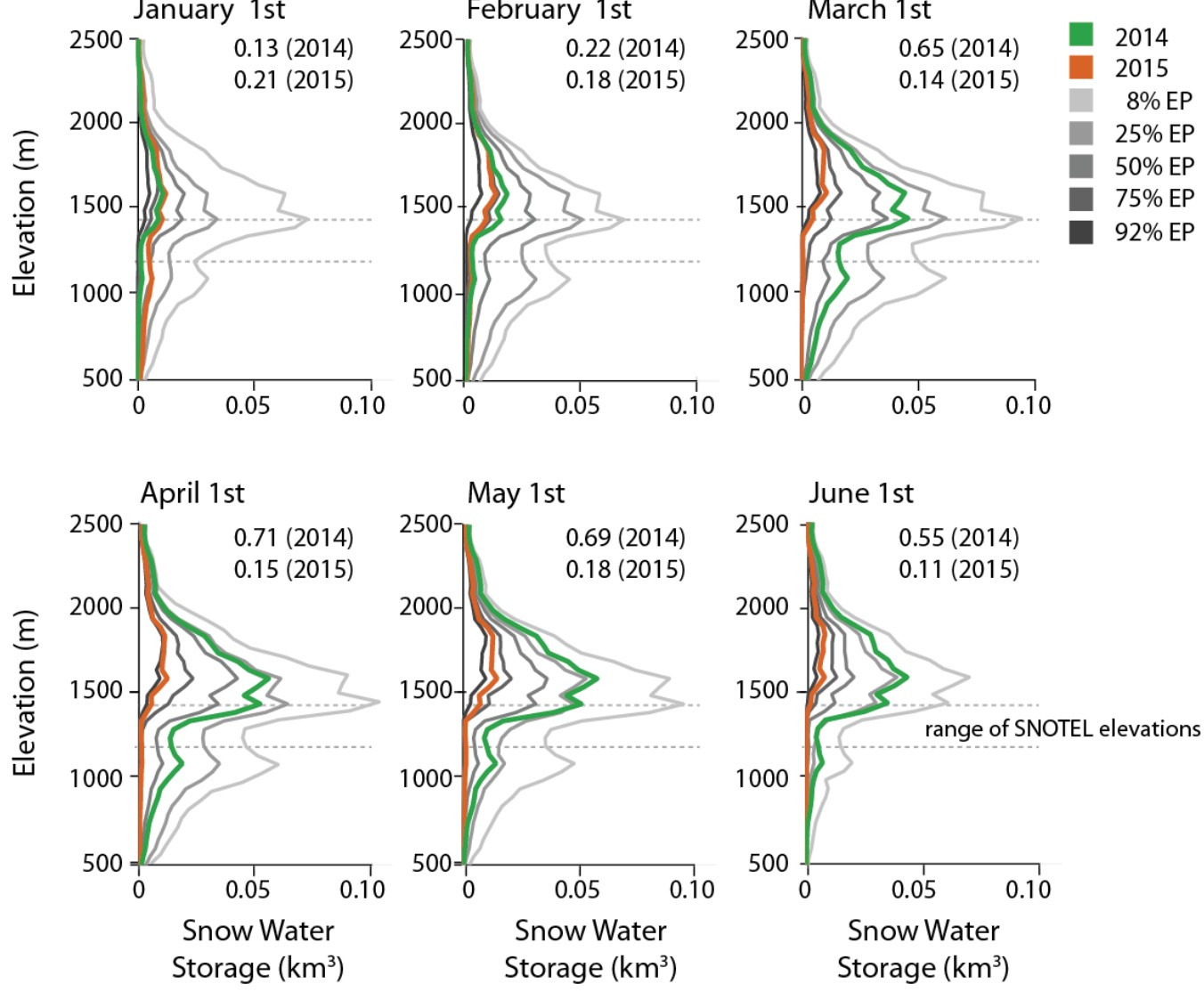

**Figure 4: Volumetric snow water storage binned by 50 m elevation bands. The corresponding basin-wide snow water storage (km³) for 2014 and 2015 is provided for each month. Larger snowpacks (lower exceedance probability) have considerable contributions at between 1000 – 1300 m. During 2014 and 2015, this elevation range had minimal snowpack, despite close to normal precipitation. Note that on the vertical axes, snow water storage below 500 m and above 2500 m are not included for visual clarity. These elevations contribute minimally to basin-wide snow water storage. The calculations are 5-day averages centered on the first day of each month.**

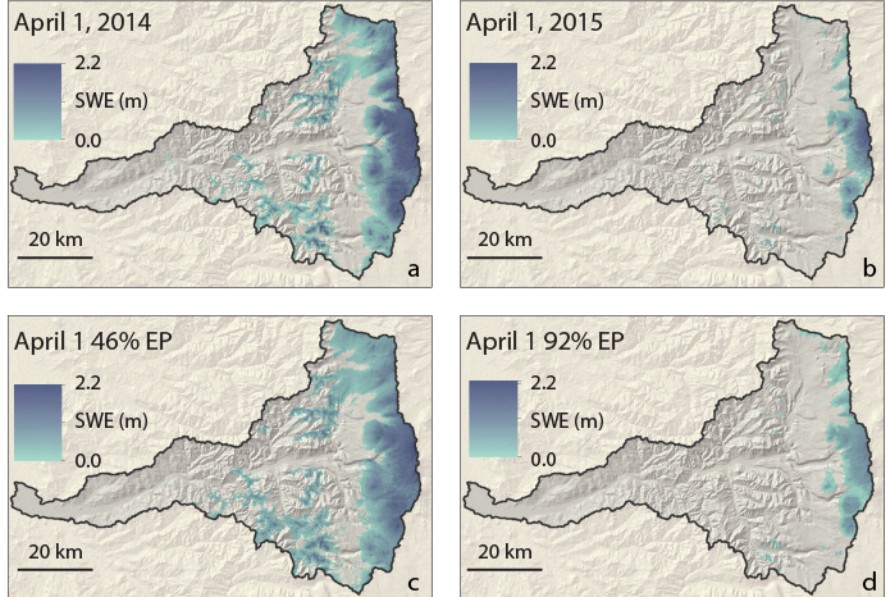

**Figure 5: The spatial distribution of SWE on April 1<sup>st</sup> from water years 2014 and 2015 as compared to the corresponding EP. Both the distribution and magnitude of SWE are strikingly similar. The calculations are 5-day averages centered on the first day of each month.**

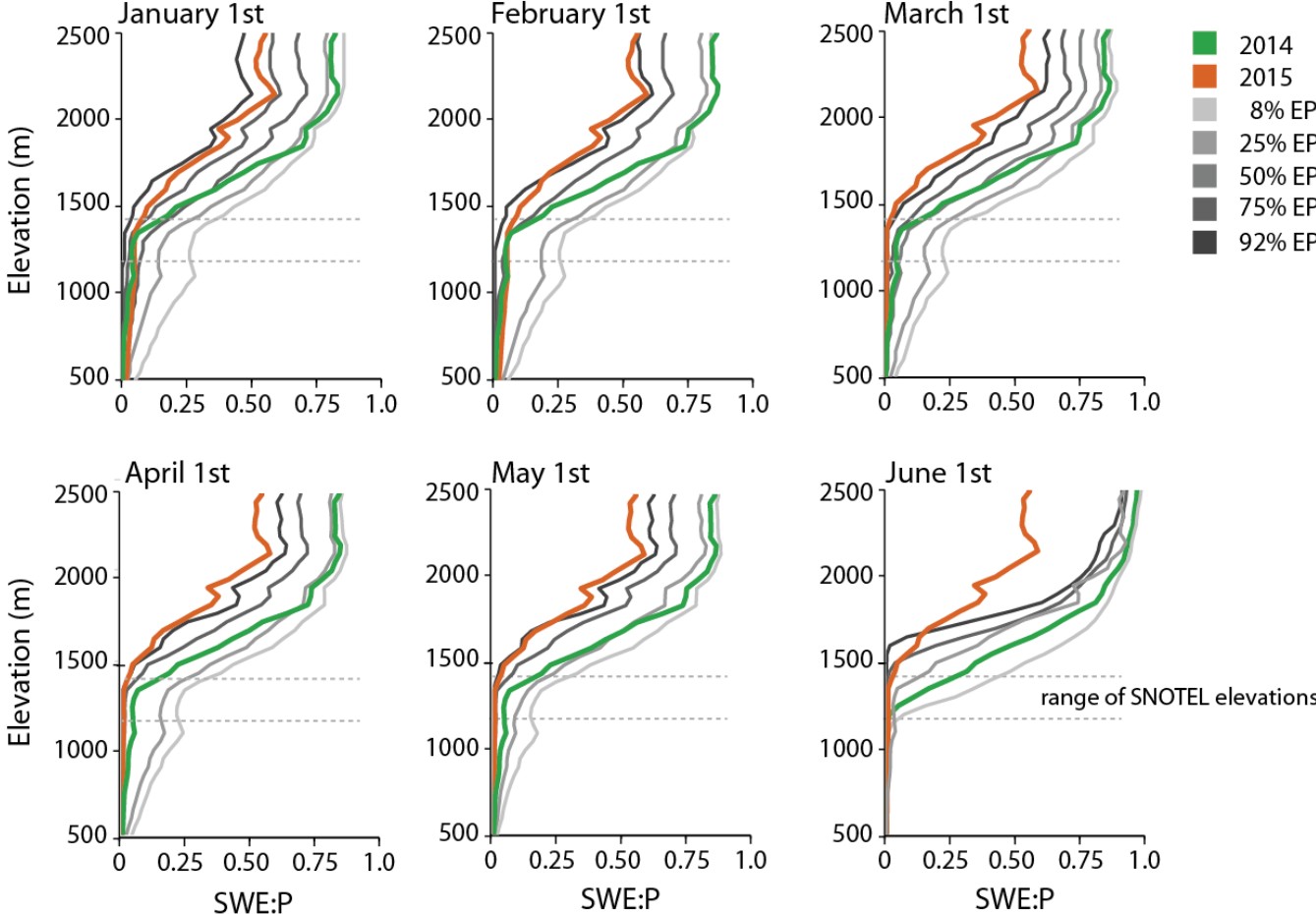

**Figure 6: The ratio of SWE:P binned by 50 m elevation bands. The relationship between elevation and SWE:P is evident across all exceedance probabilities. Under +2°C simulations and in 2014 and 2015, roughly 1500 m is the elevation at which SWE:P begins to increase substantially along the horizontal axis. Note that on the vertical axes, snow water storage below 500 m and above 2500 m are not included for visual clarity. These elevations contribute minimally to basin-wide snow water storage. The calculations are 5-day averages centered on the first day of each month.**

# Future Snow? A Spatial-Probabilistic Assessment of the Extraordinarily Low Snowpacks of 2014 and 2015 in the Oregon Cascades

Eric A. Sproles[1,2], Travis R. Roth[2], Anne W. Nolin[2]

[1]Centro de Estudios Avanzados en Zonas Áridas, Universidad de La Serena, Raul Bitran 1305, La Serena, Chile
[2]College of College of Earth, Ocean, and Atmospheric Sciences; Oregon State University, 104 CEOAS Admin Bldg, Corvallis, OR, 97331-5503, USA

*Correspondence to*: Eric A. Sproles (eric.sproles@gmail.com)

**Abstract.** In the Pacific Northwest, USA, the extraordinarily low snowpacks of winters 2013–2014 and 2014–2015 stressed regional water resources and the social-environmental system. We introduce two new approaches to better understand how seasonal snow water storage during these two winters would compare to snow water storage under warmer climate conditions. The first approach calculates a spatial-probabilistic metric representing the likelihood that snow water storage of 2013–2014 and 2014–2015 would occur under +2ºC perturbed climate conditions. We computed snow water storage (basin-wide and across elevations), and the ratio of snow water equivalent to cumulative precipitation (across elevations) for the McKenzie River basin (3 041 km[2]), a major tributary to the Willamette River in Oregon, USA. We applied these computations to calculate the occurrence probability for similarly low snow water storage under climate warming. Results suggest that, relative to +2ºC conditions, basin-wide snow water storage during winter 2013–2014 would be above average while that of winter 2014–2015 would be far below average. April 1 snow water storage corresponds to a 42% (2013–2014) and 92% (2014–2015) probability of being met or exceeded in any given year. The second approach introduces the concept of snow analogs to improve the anticipatory capacity of climate change impacts on snow derived water resources. The use of a spatial-probabilistic approach and snow analogs provide new methods of assessing basin-wide snow water storage in a non-stationary climate, and are readily applicable in other snow dominated watersheds.

## 1 Introduction

In the Pacific Northwest (PNW), mountain snowpacks during the winters of 2013–2014 and 2014–2015 were at or near record lows and well below 50% of the historic median value (National Resource Conservation Service, 2014, 2015b). For several decades the Natural Resources Conservation Service (NRCS) Snowpack Telemetry (SNOTEL) network has provided measurements of snow water equivalent (SWE; the amount of water contained within the snowpack) and meteorological data. These station-based measurements have historically served as a proxy for basin-wide snow storage and provide an effective SWE index for estimating streamflow, however under a shifting climate these statistical relationships have also

changed (Montoya et al., 2014). The PNW's extreme low snowpacks and subsequent snow water storage of 2013–2014 and 2014–2015 highlight the limitations of location-specific measurements in a shifting climate.

On March 1, 2015, 47% of snow monitoring sites in the Willamette River Basin (WRB, 29 730 km$^2$, Fig. 1) registered zero SWE while snow was still present at higher elevations. The absence of snow during the winter of 2014–2015 stands in contrast to cumulative winter precipitation, which was at 83% of normal (778 mm) for November to February (derived from PRISM data (Daly et al., 2008). While the concurrent drought in California received substantial attention, the economic and environmental impacts in the PNW were also profound. These two extreme low snowpacks in the PNW led to ski area closures, recreation restrictions, municipal water limitations, severe wildfires, low streamflows, nearly dry reservoirs, harmful algal blooms, and high fish mortality (Associated Press, 2015; Bend Bulletin, 2015; Oregon Department of Fish and Wildlife, 2015; The Oregonian, 2015a; The Oregonian, 2015b). These types of externalities highlight the importance of mountain snow water storage and the implications of snow drought.

Mountain snow water storage in the western Oregon Cascades and across the western United States serves as vital inter-seasonal storage from cool, wet winters with low water demand to hot, dry summers when demand peaks (Oregon Water Supply and Conservation Initiative, 2008; United States Army Corps of Engineers, 2001). The western Oregon Cascades form the eastern boundary of the WRB (Fig.1), and abundant winter precipitation falling in these mountains (up to 3000 mm yr$^{-1}$) sustains the 13$^{th}$ highest streamflow in the conterminous United States (Hulse et al., 2002). Even in such a wet place, snowmelt is critically important. Brooks et al. (2012) estimated that over 60–80% of summer base flow in the Willamette River derives from the snow zone at elevations over 1200 m, though this elevational zone represents only 12% of the land area and 15.6% of the annual precipitation in the basin.

The McKenzie River Basin (MRB, 3 041 km$^2$) is a major tributary to the WRB (Fig. 1), and is located in the main part of the Willamette's "at-risk" snow zone (Nolin and Daly, 2006). Snowmelt in MRB is critical to meeting environmental and societal demands of the WRB, supplying almost 25% of the river's summer discharge at its confluence with the Columbia River near Portland, Oregon (Hulse et al., 2002), despite only occupying 10% of its area. The hypsometry of the MRB and WRB are visually similar (Fig 1b) and statistically similar when tested using a two-parameter Kolmogorov-Smirnov test for sample distribution (Young, 1977).

The maritime snowpacks of the MRB, WRB, and the PNW are deep (>1.5 m), relatively warm (Sturm et al., 1995), and SWE typically reaches its basin-wide maximum on approximately April 1 (Serreze et al., 1999; Stewart et al., 2004). Nolin and Daly (2006) identified snow in the WRB as climatologically "at-risk" since it typically accumulates at close to 0ºC and can convert to rainfall with just a slight increase in temperature. As a result of changes in circulation patterns and warmer temperatures there have been declines in April 1 SWE in the PNW (Barnett et al., 2005; Kapnick and Hall, 2012; Luce and Holden, 2009; Mote, 2006; Mote et al., 2005; Service, 2004; Stoelinga et al., 2010), and peak streamflow has shifted to earlier in the year (Fritze et al., 2011; Stewart, 2009).

These shifts in streamflow highlight the challenges of using location specific measurements of SWE for prediction in changing climate. While SNOTEL sites provide valuable and robust data, they typically occupy a limited elevation range

that leads to an under-sampling of both the high elevation snow zone and the lower elevation rain-snow transition zone (Molotch and Bales, 2006; Montoya et al., 2014; Nolin, 2012). This limited range holds true in the MRB, where the mean elevation is 1424 m and the elevational range between the five stations is only 245 m.

Elevational shifts in snowpack accumulation due to observed temperature increases make the past less representative of the future (Dozier, 2011; Milly et al., 2008). Additionally, patterns of snow accumulation and melt in the PNW vary as non-linear functions of elevation, slope, aspect, and landcover (Tennant et al., 2015). Augmenting point-based measurements of SWE with metrics that effectively estimate snow water storage in a mountain landscape would include calculations for normal and extreme years across elevations and at the basin scale—especially under current climate trends (Dozier, 2011).

The dimensionless ratio of SWE to precipitation (SWE:P) represents the proportion of snow water equivalent relative to cumulative precipitation (snowfall plus rainfall) over a specified time interval (Serreze et al., 1999). This ratio normalizes snow water storage by cumulative precipitation, emphasizing the impacts of temperature on snowpack accumulation and melt. When computed for April 1, the time of year when maximum basin-scale SWE is considered to occur, this ratio can be an effective measure of the stages of accumulation and melt (Clow, 2010).

Understanding how relationships between snowpack, precipitation, and temperature will be expressed at the basin scale is particularly important in the maritime PNW. Physically-based modelling studies of climate impacts in the PNW describe reduced snow water storage and earlier streamflow across the region (Elsner et al., 2010; Hamlet, 2011; Sproles et al., 2013). These deterministic approaches provide a range of outputs of past and future conditions. However these approaches stop short of an analog approach that links an individual year from the past, particularly a low snow year, to projected conditions. Climate analogs serve as a useful device to examine potential impacts on societally relevant of predictands (e.g. forest health, environmental flows, municipal water supply), and applies previous conditions to represent potential future conditions (Hallegatte et al., 2007; McLeman and Hunter, 2010; Ramírez-Villegas et al., 2011; Webb et al., 2013). For example, Ramírez-Villegas et al. (2011) developed analogs of climate and agricultural practices to identify prior climatic events that may provide insights into the impacts of future climate change in both time and space.

Incorporating an analog approach allows planners and managers to develop anticipatory capacity, the ability to better anticipate changing scenarios as needs and context change over time (Nelson et al., 2008; Rhodes and Ross, 2009). Using the extreme low snow water storage of 2014-2015 as an example, residents of the Willamette Valley raised concerns regarding the safety and taste of domestic drinking water during the summer months. These changes in water characteristics led public works departments to examine future strategies and equipment to mitigate future water quality concerns (Hall, 2015). From a hydrological perspective, this same analog approach is also used in describing streamflow, and is most commonly framed using statistical metrics. For example, the spatial extent for a previous 100-year flood event serves as an analog of floodplain dynamics and provides anticipatory capacity for land use planners and water managers.

Based on the premise that future snow water storage conditions will resemble previous winters that were warm, Luce et al. (2014) developed spatial and temporal analogs of snow water storage sensitivity to temperature and precipitation across the western United States using point-based SNOTEL data. Similarly, Cooper et al. (2016) applied model-based analyses to

compare the winters of 2014 and 2015 to projected future conditions using individual metrics of snowpack (Snow Disappearance Date, Date of Peak SWE, and Duration of Snow Cover) at SNOTEL locations in the Oregon Cascades. This approach is informative, even though point-based analysis in projected warmer conditions may not represent basin-wide conditions (Dozier, 2011; Milly et al., 2008), specifically as the rain-snow transition shifts towards higher elevations (Nolin et al., 2012; Nolin and Daly, 2006).

To develop statistically valid analogs for snow water storage and snow water storage at the basin scale requires a spatially explicit, probabilistic approach that calculate the statistical likelihood of SWE across a topographically complex mountain basin. For example, to address the question "What is the likelihood that the snow droughts of WYs 2014 and 2015 will occur in the future?" can be addressed by developing statistical thresholds of SWE and SWE:P with regards to time and location. This spatial-probabilistic approach develops upper or lower limits of predicted snow water storage conditions throughout a watershed. While probabilistic approaches are common to streamflow hydrology, spatial approaches to probabilistic questions are less common. A notable application of a spatial-based, probabilistic approach was developed by Graf (1984). This research applied 107 years of channel migration records to calculate the probability of subsequent erosion in a given parcel, creating a probabilistic map of river movement. The map outlined the character of the river system that identified areas where channel migration was more likely to occur. Margulis et al. (2016) characterized the extreme California snow deficit of 2015, but did not compare this snow drought to potentially warmer climatic conditions. Snow hydrology models can readily incorporate climate change projections (Adam et al., 2009; Sproles et al., 2013) and model outputs can be assessed using a spatial-probabilistic framework that explicitly accounts for elevation.

This research introduces a physically based, spatial-probabilistic modelling framework to compare the extraordinarily low snow winters of WY 2014 and WY 2015 (WY=Water Year, defined as 1 October – 30 September in the western United States) in the context of warmer climatic conditions. Our approach captures the spatial variability of mountain snow water storage under warmer temperatures across decades by simulating the variability of SWE and SWE:P at the basin scale for 23 WYs using +2ºC conditions. These outputs are used to frame the snow water storage of WY 2014 and WY 2015 in the context of future snow and snow analogs. This approach is intended to build anticipatory capacity for climate change impacts in the PNW through snow analogs. While limited to the McKenzie River Basin (a well-studied watershed that is characteristic for maritime snow in the WRB (Nolin and Daly, 2006), regional sensitivity to climate warming makes PNW snowpack and snow water storage, and those in similar maritime climates, acutely vulnerable to snow drought (Leibowitz et al., 2014; Nolin and Daly, 2006).

Specifically, we ask:

- How does snow water storage from WY 2014 and WY 2015 compare to snow water storage under +2ºC conditions?
- What is the probability that similar snowpacks and snow water storage will occur in the future?
- How does snow water storage during WY 2014 and WY 2015 vary by elevation?

**2 Research Methods**

Our approach applies a spatially-distributed and physically-based snow hydrology model to compute probabilities of SWE and SWE:P for 23 WYs under +2C winter conditions. We then model WY 2014 and WY 2015 snow water storage and these outputs provide probabilistic context for the snow water storage of those two winters. Below we provide details on the study area and specific methods used in this approach.

This study focuses on the McKenzie River Basin. In addition to the MRB being a major tributary to the Willamette River, it has a well-developed network of meteorological stations associated with the HJ Andrews Long Term Ecological Research site, four SNOTEL stations, four dams for flood control and hydropower, serves as the primary source of domestic water for 200 000 people, and is home to federally protected salmonids, amphibians, and mussels. The MRB is characterized by wet winters and dry summers, with average annual precipitation ranging from 1000 mm to 3000 mm that follows the elevation gradient (114–3147 m). Elevations between 1000 and 2000 m comprise 42% of the MRB's total area (Fig. 1a) and 93% of the total snow water storage in the MRB (Sproles et al., 2013). While elevations above 2000 m accumulate the most SWE per unit area, that zone comprises only 1% of total area and 6% of the total snow water storage for the MRB. In terms of volume, snow is the primary seasonal water storage mechanism in the MRB with historic mean basin-wide snow water storage (SWE × area; 1989–2009) of 1.26 km$^3$ on April 1 (Sproles et al., 2013), compared with total reservoir storage of 0.40 km$^3$ (United States Army Corps of Engineers, 2016; United States Department of Agriculture, 2016). By comparison groundwater storage for the MRB was estimated to be roughly 4 km$^3$, with a mean transit time of seven years (Jefferson et al., 2006).

Spatially-distributed values of precipitation and SWE were computed using SnowModel (Liston and Elder, 2006a, 2006b) for WY 1989–2012. SnowModel is a spatially distributed, process based model that computes temperature, precipitation, and the full winter season evolution of SWE including accumulation, canopy interception, wind redistribution, sublimation, evaporation, and melt. The model framework applied in this study is the same as applied in Sproles et al. (2013), with the addition of a multi-layer snowpack algorithm. Because the modelling framework is physically-based and spatially-distributed, perturbations to temperature inputs will propagate throughout the model including absolute humidity and energy balance calculations, thus maintaining the dependencies between snowpack and temperature. WY 2005 was excluded due to prolonged regional temperature inversions that were not resolved in the model (Sproles et al., 2013).

Model input data were derived from SNOTEL and station data within the study area (six stations in total), nearly spanning the full elevation range of the MRB (Fig. 1; Sproles et al., 2013). The 23-year set of model forcing data includes winters with above average, normal, and below average snowpack; positive, negative, and neutral ENSO climate patterns; and cool and warm phases of the Pacific Decadal Oscillation (Brown and Kipfmueller, 2012). The model was run at a daily time step and 100-m grid resolution. In the calibration and validation phase, the model was first calibrated to temperature and precipitation to ensure that the model results were representative of these first order inputs, with mean Nash-Sutcliffe Efficiencies (Legates and McCabe, 1999; Nash and Sutcliffe, 1970) of 0.80 and 0.97 respectively. The model was then calibrated for

physical snowpack conditions (mean Nash-Sutcliffe efficiency of 0.83 for automated stations, 0.70 for field locations, and an overall spatial accuracy of 82% compared with Landast fSCA data). For a detailed description of the model structure, calibration, validation, and performance please refer to Sproles et al. (2013).

Using the validated model, we increased temperatures by +2ºC and re-ran the model over the same timeframe and spatial domain. Projections for future precipitation in the WRB and the PNW are highly uncertain (Safeeq et al., 2016), and in the Oregon Cascades temperature, not precipitation, dominates the accumulation and melt cycles of snowpack (Sproles, 2012; Sproles et al., 2013). Our delta increase to temperature is intended to be straightforward, and to avoid the uncertainties associated with precipitation in this region.

We extracted SWE and precipitation (P) data, and computed 5-day averages for each centered on the first day of each month for January to June, for every year in the model run, and for each grid cell in the model domain. These 5-day mean values were used to minimize any effects from individual events (melt, snowfall) while still capturing the overall snow water storage characteristics at the beginning of the month.

Exceedance probability (EP) is a widely used hydrologic metric describing the statistical likelihood that a value of a given magnitude or greater will occur in a specified time period (e.g. annually) (Sadovský et al., 2012; Salas and Obeysekera, 2013). Expressed as a percentage, it is calculated as:

$$EP = \left(\frac{m}{n+1}\right) \times 100 \qquad (1)$$

where, $m$ is the rank of the data value (ranked from highest to lowest) and $n$ is the total number of data values (Dingman, 2002).

For example, 20% EP (a low annual exceedance probability) is the statistical likelihood that a value could be met or exceeded 20% of the time, or a 1 in 5 chance of occurring or being exceeded in any year. 20% EP represents a relatively large value. A 90% EP (a high annual exceedance probability) describes the statistical likelihood of a measurement that would be met or exceeded in 90% of the time, and represents a relatively low value. EP is commonly applied to point-based data such as a stream gage or SNOTEL station. However, because mountain snow water storage varies by elevation, slope, aspect, and landcover (Tennant et al., 2015), we expanded point-based EP calculations to the watershed scale to include normal and extreme years.

To accomplish a spatial perspective of exceedance probability, we applied 23 years of model output to compute the EP for the first of the month (January to June) based upon the 5-day averaged SWE and SWE:P values for each grid cell in the model domain. The dimensions of the model domain is a grid of 759 rows × 1121 columns. In order to sort each grid cell individually across the 23 datasets (years), the two-dimensional data sets (759 rows × 1121 columns) was decomposed into 23 one-dimensional vectors (1 × 850,839) then combined to create a 23 × 850,839 matrix. The location information of each grid cell was retained for subsequent mapping and analysis. For each year, the 23 values in each row were sorted from

highest to lowest. The 23 × 850,839 data matrix was recomposed into 23 data matrices of dimension 759 × 1121 creating a corresponding spatial exceedance probability matrix. This was completed for each month (January to June).

To respond to the question "How do snow water storage from WY 2014 and WY 2015 compare to snow water storage under a warmer climate?" we modeled SWE and SWE:P using SnowModel with meteorological forcing data from WY 2014 and WY 2015 for the MRB, with the same stations as from our previously validated model runs. These model runs were also validated using the same methods as described in Sproles et al. (2013). We then compared the snowpack metrics from these two winters with model output from the +2°C climate scenario.

Elevation is the most important physiographic variable in determining SWE in this basin (Nolin, 2012) so we aggregated the data into 50-m elevation bands (Fig.1a). In each of these bands we computed snow water storage ($km^3$) and mean SWE:P (m/m). This allowed us to understand the variation of snowpack properties by elevation, their spatial probability of occurrence, and the statistical context for the extraordinary snowpacks of WY 2014 and WY 2015.

*An important point to bear in mind is that the EP values were computed using perturbed meteorological forcing data (+2°C), while values for WY 2014 and WY 2015 were derived from unperturbed meteorological forcing data.*

## 3 Results

For context, historically in the MRB, 62% of precipitation falls in the Nov to Mar (N–M) time period as calculated from monthly precipitation data from 30-year PRISM gridded climate normals (Daly et al., 2008). Within that period, Dec to Feb (DJF) are historically the coldest and wettest months (Daly et al., 2008). For N–M in WY 2014, precipitation was at 102% of the 30-year normal (calculated from PRISM data) and temperatures at SNOTEL stations in the MRB were 0.9°C warmer than normal (National Resource Conservation Service, 2015a). For the DJF period, WY 2014 monthly precipitation was 96% of normal and SNOTEL temperatures were 0.7°C warmer than normal. During WY 2015, N–M precipitation was 81% of the 30-year average but temperatures in the snow zone were 2.7°C warmer than average. For the DJF period of WY 2015, monthly precipitation was 78% of normal and temperatures in the snow zone were 3.3°C warmer than normal (National Resource Conservation Service, 2015a). To provide historical context, Fig. 2a and b presents graphically presents the 30-year precipitation and temperature normals from the PRISM datasets as compared to WYs 2014 and 2015. Fig. 2c presents modeled basin-wide snow water storage for WYs 2014 and 2015 as compared to the 23-year mean from the +2°C snowpack simulations.

A warmer than normal Jan 2014 limited snowpack accumulation during the early portion of the winter, and wetter than normal conditions in Feb 2014 accompanied by near normal mean temperatures increased basin-wide snow water storage to near average/above average snowpack conditions (as compared to a +2°C perturbation) for the remainder of the season (Fig. 2c). The warmer than normal conditions that persisted throughout WY2015 greatly inhibited seasonal snowpack accumulation, despite above average precipitation in Mar 2015 (Fig. 2c). For a more detailed long-term climate analysis please refer to Abatzoglou et al. (2014).

**3.1 Snow Water Storage**

In the context of our exceedance probability framework, we see that the April 1 basin-wide snow water storage for WY 2014 falls between the 42 and 46% EP, meaning that WY 2014 snow water storage is slightly above average for a +2ºC model perturbation (Figs. 2, 3, 4, and 5a and c). Snowfall occurring after April 1, 2014 improved late season snow water storage corresponding to 33% and 25% EP for May and June, respectively (Figs. 3 and 4). In WY 2015 basin-wide snow water storage was well below historical conditions, even when compared with +2ºC conditions. April 1 snow water storage for WY 2015 corresponds to 92% EP (Figs. 3, 4, 5b, and 5d). In that year, there was little late spring snowfall, so unlike WY 2014 basin-wide snow water storage did not increase (Fig. 3). WY 2015 was also notable in that peak snow water storage occurred in January and was only 0.21 km$^3$, corresponding to 79% EP (Figs. 3 and 4).

Fig. 4 shows the spatial exceedance probabilities for the +2ºC model runs, aggregated into 50-m elevation increments (WY2014, 42% EP; WY2015, 92% EP). For most years, the total amount of April 1 snow water storage is greatest within the elevation range of 1300–1800 m. However in WY 2015 this mid-elevation zone (1300–1800 m), representing 393 km$^2$ (as calculated from the elevation dataset) is essentially snow-free (Fig. 4). Snow water storage in this elevation range is critical for late season runoff, as 1200 m represents the elevation threshold for summer baseflow contributions (Brooks et al., 2012). From a spatial perspective, Fig. 5 presents the distribution of SWE in the MRB in WYs 2014 and 2015 on April 1, as compared to the 46% and 92% EP (as compared to a +2ºC perturbation), respectively. These figures show snow water storage is almost entirely limited to the upper portions of the basin, and that the more spatially extensive mid-elevations where snow accumulates historically are snow free. In other words, in WY 2014 and 2015 the zone where snowmelt has historically contributed most to groundwater recharge (Jefferson et al., 2008; Tague and Grant, 2009), shifted to rain. Jefferson et al. (2008) showed that the recharge signal varies spatially and temporally, and that the location of the rain-snow transition is the dominant control on recharge for at the watershed scale.

**3.2 SWE:P**

This elevation dependent shift from rain to snow is evident in Fig. 6, where at an elevation of 1200 m, SWE:P is below 0.06 for the period January to June in both WY 2014 and 2015. This ratio does not exceed 0.20 until an elevation of 1500 m in WY 2014, which is still markedly lower than the long-term mean SWE:P at the McKenzie SNOTEL site (0.58, 1454 m). In WY 2015 this 0.20 threshold is not reached until an elevation of 1750 m, approximately 300 m above the highest elevation SNOTEL site in the MRB, and thus was not captured in the SNOTEL data. From February to May in WY 2014, SWE:P increases due to late season storms that added snow water storage, and remained above 50% EP when compared with +2ºC conditions. From February to May in WY 2015, SWE:P never surpasses the 0.60 threshold, and remains below 90% EP when compared with +2ºC conditions.

## 4 Discussion and Conclusion

The winters of 2014 and 2015 had very low snowpacks across the Pacific Northwest due to higher than normal winter temperatures but average or near-average precipitation (Fig. 2, *National Resource Conservation Service*, 2014, 2015b), highlighting the sensitivity of the region's snowpack to increased temperature. In the MRB snow zone mean temperatures (Nov-Mar) were 0.9ºC above the 30-year normal in WY 2014, while WY 2015 were 2.7ºC above normal. These low snow years persisted even under normal and slightly below normal Nov-Mar precipitation (WY2014, 102%; WY2015 81%). The SWE:P metric also identifies increased temperature, rather than reduced precipitation, as the primary reason for the diminished snow water storage of WY 2014 and WY 2015, especially at mid elevations. At 1500 m the April SWE:P values for the two years are considerably different (Fig. 6; WY2014, SWE:P = 0.22, 60% EP; WY2015, SWE:P = 0.04, 95% EP). As such, these two winters' extraordinarily low snowpacks offer an analog perspective for projected future snow conditions in the MRB and potentially the Willamette River Basin. WY 2014 serves as a snow analog for slightly warmer conditions (+1ºC), with an EP between 42 and 46%. While WY 2015 serves as a snow analog for conditions increasing beyond 2.5ºC with an EP of around 92%. The volumetric difference between the two years is considerable (0.56 km³), representing 1.4 times more than the total reservoir storage capacity of the MRB (United States Army Corps of Engineers, 2016; United States Department of Agriculture, 2016).

The SWE:P metrics across elevation bands provides a simple yet telling description of precipitation phase (rainfall vs. snowfall) and evolution of snow water storage (accumulation and ablation). The shifts from rain to snow seen in the modelled results highlight the limitations of a monitoring network that occupies a limited range. In the MRB the SNOTEL stations occupy a mean elevation of 1424 m with a range of only 245 m. During WY 2014 and 2015, this limited range did not capture zones with maximum snow water volume and were essentially below the rain-snow transition (Figs. 4 and 6). This same under representation of snowpack was found throughout the greater WRB with 47% of snow monitoring sites registering zero SWE while snow was still present at higher elevations on March 1, 2015.

As precipitation shifts from snow to rain, the SWE:P metric can augment individual values of SWE and P to provide key information on shifts in water storage throughout the course of a winter, and valuable insights to water resource managers in a non-stationary climate. For example, on March 1, when basin wide SWE is typically approaching its maximum, both years are essentially snow free at 1200 m. A low SWE:P ratio in March under normal winter precipitation conditions could indicate peak streamflow has occurred or most likely would occur earlier in the year, which has important implications for water resource management in subsequent months.

Low snow water storage and shifts in streamflow negatively impact water quantity, water quality, hydropower operations, winter snow sports, and summer recreation. In WY 2015, record low snow water storage led to summer drought declarations, extreme fire danger, and modified hydropower operations in the MRB. The typical consistent flow of the groundwater-fed McKenzie River was at 63% of August-September median flow (United States Geological Survey, 2015). Hoodoo Ski Area,

located at Santiam Pass, was open for only a few weekends in WY 2014 and in WY 2015 they suspended operations in mid-January, the shortest season in their 77-year history. In the adjacent Santiam River Basin (north of the MRB), diminished snow water storage and less-than-anticipated spring rains in WY 2015 pushed the Detroit Reservoir (storage capacity 0.35 $km^3$) to historic low levels. In May harmful blue-green algae concentrations were above acceptable amounts by seven-fold, and July reservoir levels were approximately 21-m below capacity. Concerns over the taste and safety of domestic drinking water in the Willamette Valley prompted municipal water managers to explore options for upgrading water treatment facilities.

At more broad timescales the shift from snow to rain at mid-elevations could also potentially impact groundwater recharge. The rain-snow transition is the dominant control on recharge in the MRB, and varies spatially and temporally (Jefferson et al. 2008). Because groundwater storage is large and transit times in the MRB are approximately 7 years (Jefferson et al. 2008), the full impacts of WY2014 and WY2015 on ground and surface water resources are not yet known.

Water quality, energy production, and recreation externalities are not well represented in deterministic models, but become challenging realities that the public faces in years with low snow. Intervention strategies can fail because they lack adequate information about the impacts of climate change that are not incorporated into deterministic physical models and play out at the human scale (Ramírez-Villegas et al., 2011). Transitioning from purely deterministic approaches (i.e. snow water storage is reduced by a certain percentage) to ones that link climate and snow conditions with real world impacts provide a complementary perspective for mitigation and adaptation. Our analog approach combines projected climate impacts with the extreme low snow years of 2013-2014 and 2014-2015 for insights into improved management in shifting conditions. Such an analog approach allows planners and managers to develop adaptation and mitigation strategies that use the past to demonstrate what did or did not work under climate stress, and help build a more informed understanding of ways to improve future planning efforts (Ramírez-Villegas et al., 2011).

Climate change impacts are often expressed in probabilistic terms (Randall et al., 2007) and so it is logically consistent to estimate snowpacks and snow water storage in this manner. This research does not assume that the probabilities presented here are based upon a precise representation of future conditions nor that future climates will be +2ºC warmer every winter. We present these results as a way to frame the likelihood of future basin-wide snow water storage in the context of our current understanding of climate change. These probabilistic insights are then used to identify WY 2014 and WY 2015 as analogs years for managers and decision makers. The WY 2014 snow water storage would be slightly above average for +2ºC conditions; and the WY 2015 snow water storage would be very low snow water storage for +2ºC conditions, but not a record low. These analog years thus provide guidance for adaptation strategies to mitigate potential failures of existing management plans.

Our spatially explicit approach augments information from the existing SNOTEL network. While SNOTEL data continue to play a key role for seasonal streamflow forecasting under historic climatic conditions, these statistical relationships have been changing (Montoya et al., 2014). While providing modern scientific equipment, SNOTEL sites in the MRB occupy a limited range (245 m) in the mid-elevations and may not capture basin-wide snow water storage in warmer conditions. For

example, in the MRB all SNOTEL sites in the MRB were snow-free for most of February to March 2015 and therefore incapable of providing predictive skill for water resource management. Our basin-scale probabilistic approach provides a more complete picture of water storage and captures the elevation variability absent in point-based measurements.

The winters of WY 2014 and WY 2015 demonstrate a considerable departure from the stationary snow water storage conditions on which present-day management plans are based. With continued current warmer climates, the snow water storage conditions represented by these two winters are more likely to occur. In the meantime, the value of spatially explicit probabilistic calculations rests in the ability to better define the range of statistical outcomes of subsequent winters that are representative of basin-wide conditions. Framing the low snow water storage of WY 2014 and WY 2015 as analogs of future snow provides insights into potential climate impacts and externalities on social and environmental systems. Together, probabilistic metrics and snow water storage analogs can help build capacity to better anticipate hydrologic changes in a warming climate.

**Acknowledgements**

This research was made possible by support from the National Science Foundation (BCS-0903118 and EAR-1039192). We gratefully acknowledge the modelling guidance of Dr. Glen Liston. The data for Figs. 3, 4, and 6 can be downloaded at http://people.oregonstate.edu/~sprolese/snow_frequency/. The authors would also like to thank the two anonymous reviewers for their comments and expertise the Associate Editors of the Cryosphere for managing the submission and revision process.

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

**Figures:**

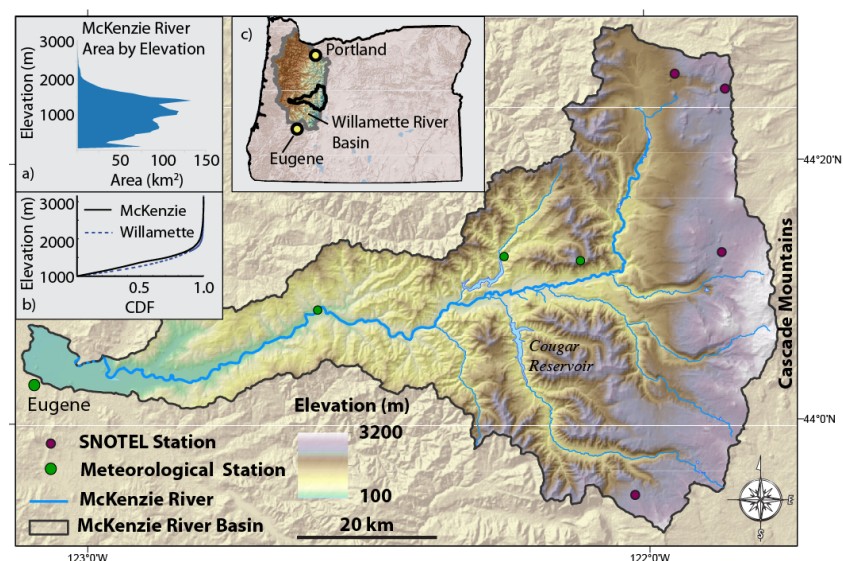

**Figure 1: Context map of the McKenzie River basin, and its geographic relationship to the Willamette River basin. The geographic locations of the SNOTEL other meteorological stations used as model forcings show the altitudinal range of inputs. Inset figure a) represents the area by elevation for the McKenzie River basin. Inset figure b) presents the Cumulative Distribution Functions (CDF) for the elevation of the Willamette and McKenzie River Basins for elevations above 1000 m, and is separated into 50m bins.**

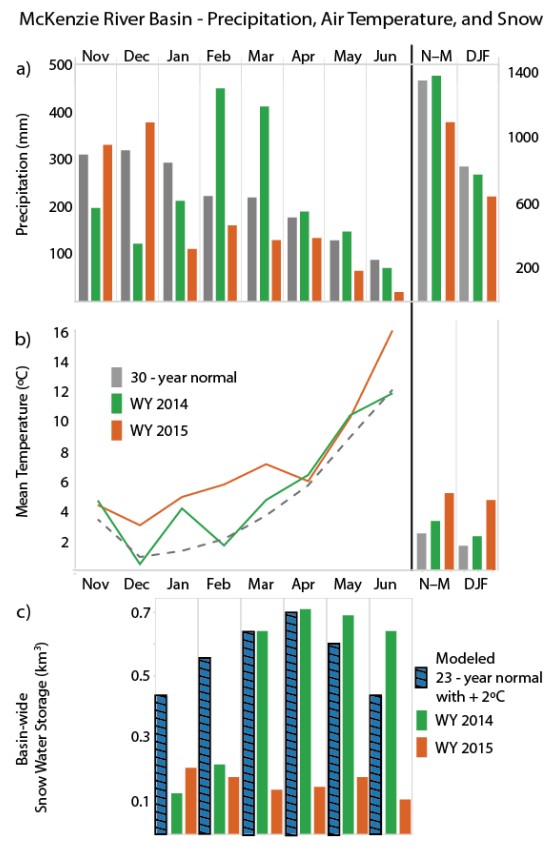

**Figure 2:** The total precipitation (a) and mean temperatures (b) for the McKenzie River basin for water years 2014 and 2015 as compared to the 30-year normal (from the PRISM datasets). The lower figure (c) represents Basin-wide Snow Water Storage for the McKenzie River Basin for water years 2014 and 2015 and the normals (+2°C) calculated from the 23 years used in this study. The calculations for snowpack are 5-day averages centered on the first day of each month.

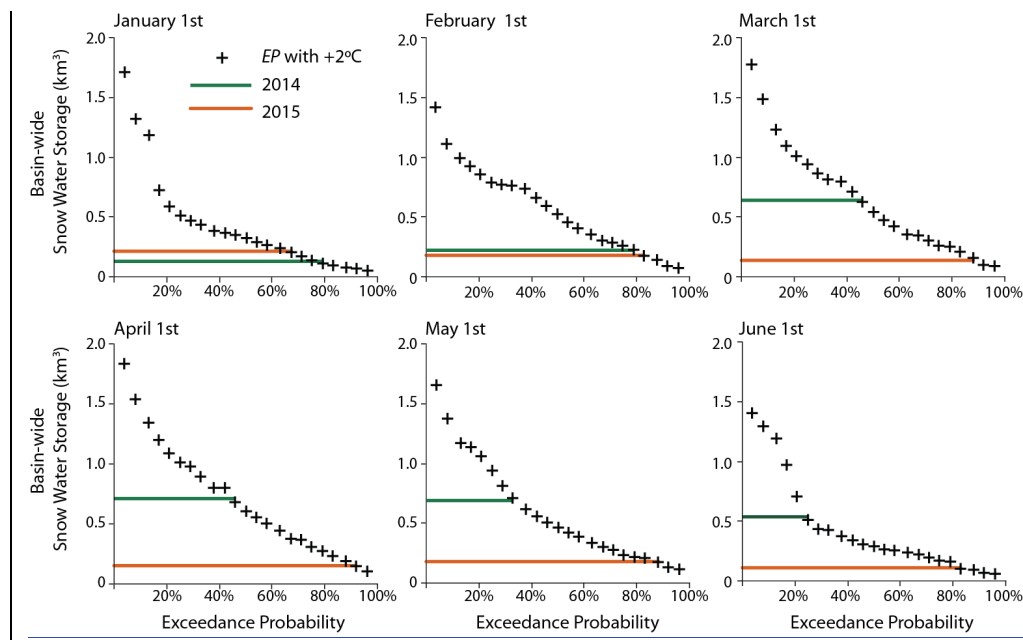

**Figure 3: The exceedance probability of basin-wide snow water storage under +2°C conditions. During 2014 snow water storage increased considerably in March to reach above average conditions. The snowpack during the winter of 2015 was extremely low, and never increased beyond 0.21 km³. The calculations are 5-day averages centered on the first day of each month.**

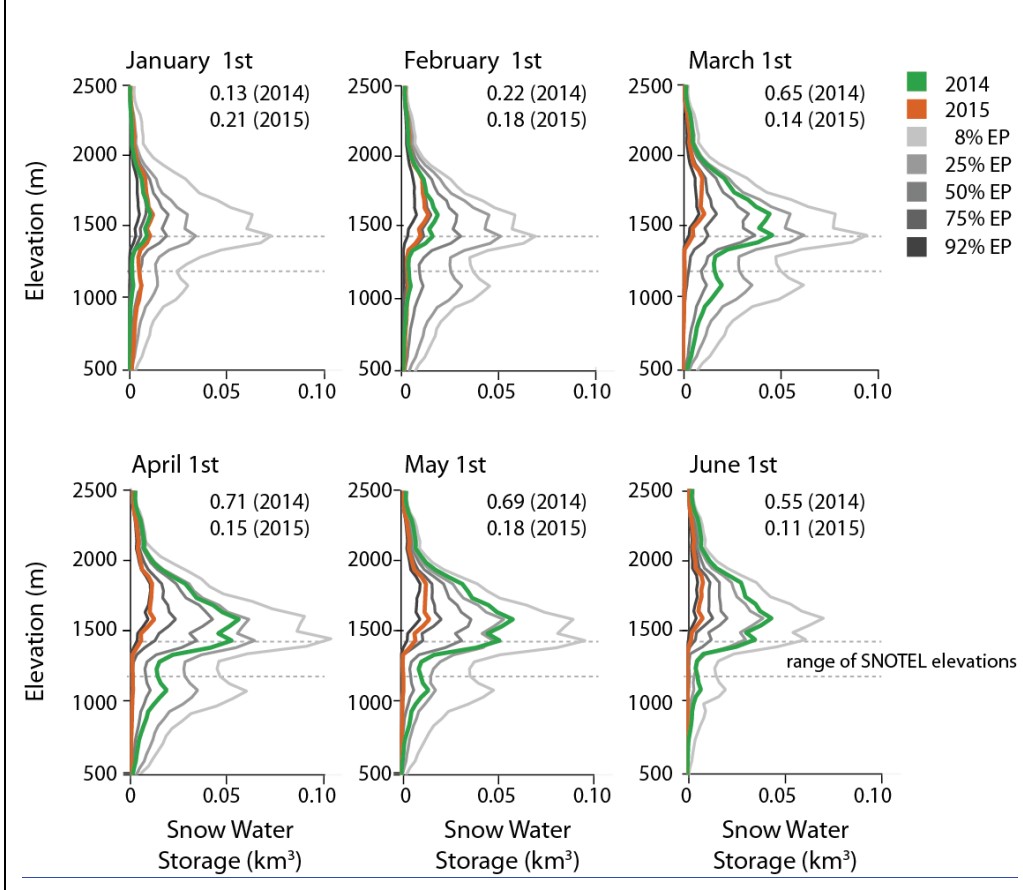

**Figure 4: Volumetric snow water storage binned by 50 m elevation bands. The corresponding basin-wide snow water storage (km³) for 2014 and 2015 is provided for each month. Larger snowpacks (lower exceedance probability) have considerable contributions at between 1000 – 1300 m. During 2014 and 2015, this elevation range had minimal snowpack, despite close to normal precipitation. Note that on the vertical axes, snow water storage below 500 m and above 2500 m are not included for visual clarity. These elevations contribute minimally to basin-wide snow water storage. The calculations are 5-day averages centered on the first day of each month.**

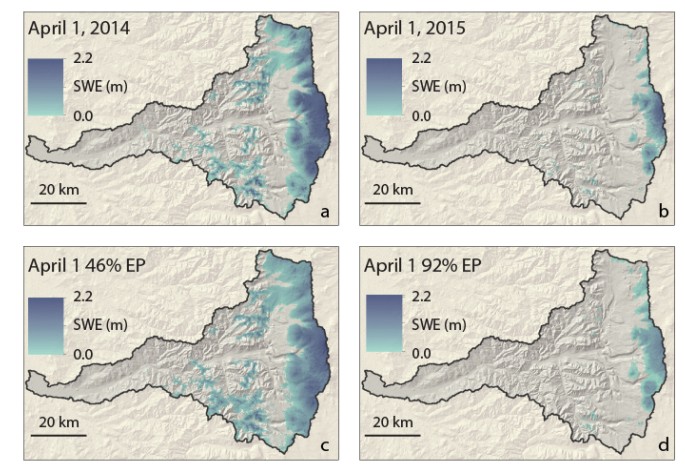

**Figure 5: The spatial distribution of SWE on April 1st from water years 2014 and 2015 as compared to the corresponding EP. Both the distribution and magnitude of SWE are strikingly similar. The calculations are 5-day averages centered on the first day of each month.**

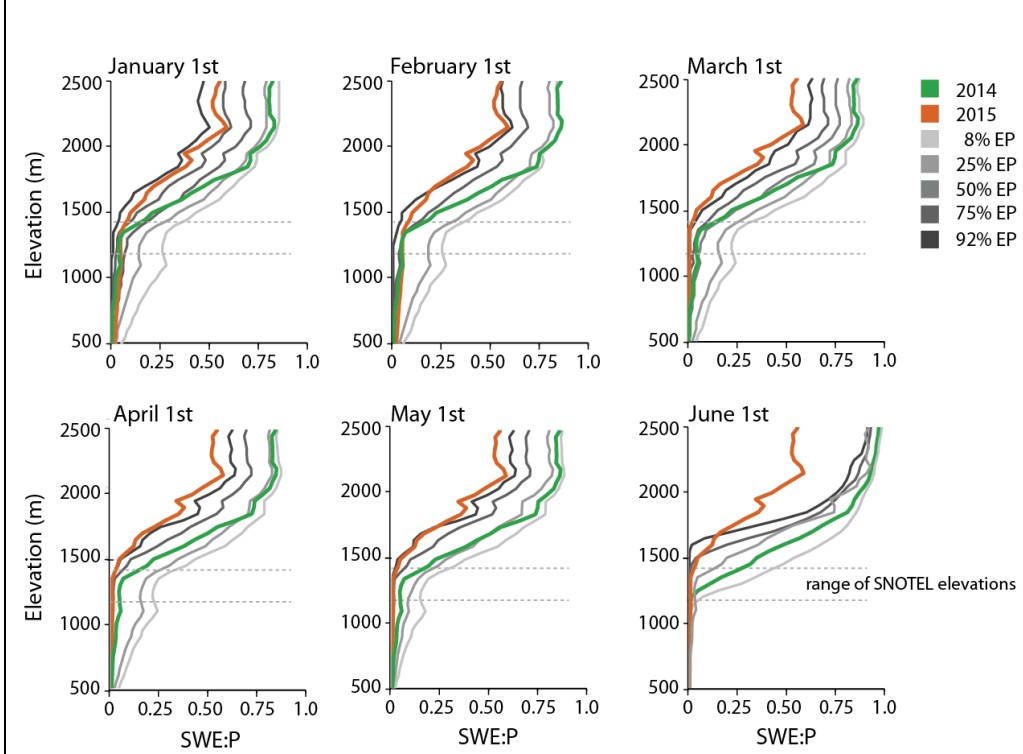

**Figure 6: The ratio of SWE:P binned by 50 m elevation bands. The relationship between elevation and SWE:P is evident across all exceedance probabilities. Under +2°C simulations and in 2014 and 2015, roughly 1500 m is the elevation at which SWE:P begins to increase substantially along the horizontal axis. Note that on the vertical axes, snow water storage below 500 m and above 2500 m are not included for visual clarity. These elevations contribute minimally to basin-wide snow water storage. The calculations are 5-day averages centered on the first day of each month.**