# Peer review of "Future Snow? A Spatial-Probabilistic Assessment of the Extraordinarily Low Snowpacks of 2014 and 2015 in the Oregon Cascades"

_The Cryosphere, 2016_

## Referee Comment (RC1) · Anonymous Referee #1 · 14 Jun 2016

This paper is an interesting and detailed look, using modeling, at the snow volume in two recent years in a small (3000 km2) river basin in the western Oregon Cascade mountain range. The most interesting finding may be that there was virtually no snow below about 1300m in 2015, yet this finding is buried in the discussion. It should be highlighted in the abstract.

big picture issues

1) Selecting 2014, 2015, and a notional year that is 2°C warmer than a 30-year "normal" presents a muddled message. In some places, 2014 is presented as being exceptional (e.g. page 1 lines 23-24); but in other places, it does not seem so exceptional (e.g. page 7 line 2 and the caption of Figure 3). It is a little puzzling why the EP would

be shown (Fig 3) for the +2°C world but not the 30-year normal. I don't know how to un-muddle this message.

2) The paper is fairly rife with unsubstantiated assertions. I mention several below.

3) spatial domain. much of the introduction refers to western US or PNW, and only on page 4 do we learn that the study area is actually a small (∼3000 km2) sub-watershed of the Willamette River Basin, and it is there asserted (without evidence) that the McKenzie River Basin is 'characteristic for maritime snow in the Pacific Northwest'. The abstract should make clear where the study was conducted, and either the attempts to relate this to the PNW should be dropped or some additional analysis should be conducted. I'm not suggesting repeating the full detailed analysis on a wider domain, but the interesting finding that there was essentially no snow below 1300m in 2015 should be easy to check with SNOTEL sites throughout the Cascades (perhaps with a latitude-elevation adjustment).

minutia:

word choice: snow volume is variously called snow water storage, snowpack storage, and just snowpack. I suggest picking a single term and using it consistently.

page 2 lines 7-9: assertions about impacts need references.

page 2 line 16 no definition of 'critical' fits this usage - the word is often, as here, misused in place of 'critically important' or just 'important'.

page 3 lines 2-3 why not? do you have evidence to back up this assertion?

page 3 lines 11-13 i don't follow the argument here. these deterministic approaches can also be (and have been) used to simulte past and future. the sentence seems to be arguing that "not analogous" is a weakness, but it's not clear why that's a weakness. likewise, the last sentence in the paragraph returns to the notion of limitations but it's not clear why. and in lines 22-23 again, it's not clear whether this assertion is also a criticism of the deterministic approach (i.e. "Only an analog approach...) I
suggest retooling the paragraphs on lines 9-29 (and possibly the next paragraph too) to better set up the strengths and weaknesses of deterministic modeling and analog approaches. Or just drop entirely.

page 3 line 34 - again, not clear whether this paragraph is describing strengths and weaknesses. the first two sentences are about analogs, but are the models referenced on line 34 the standard deterministic distributed models? Given the last 1+ page of the introduction, I was surprised to find that the paper uses a physically based snow hydrology model instead of analogs.

page 4 lines 13-14: another unsubstantiated assertion.

Page 4 line 28 "most SWE" clarify "per unit area" if that is what's meant

page 4 lines 30-32 these are really valuable and interesting comparisons. Need a source for the reservoir storage statement.

page 5 line 20-22: just call it a sensitivity test. Mote & Salathé is dated (CMIP3 vs CMIP5; see e.g. Dalton et al., Island Press 2013) and the link to IPCC is dubious, since the number discussed in the 2013 IPCC report was not exactly a "threshold set" and moreover is a global number not regional.

page 6 lines 3-10 no rationale is given for this re-dimensionalizing. Perhaps if the meaning of "spatial exceedance probability" was clear.

page 6 lines 4-5 number agreement "dimensions...has" - maybe just delete "the dimensions of"

page 7 line 11 - "is greatest" for the $+2°C$ case. Figs 4 and 5 - these are a crucial point of the paper and perhaps its main contribution. can you comment on the strengths and weaknesses of the observing network in this elevation band? is it adequate? is there evidence that these findings apply outside MRB?

page 7 line 19 - what does "subsequent runoff" mean? wouldn't snowfall also produce

"subsequent runoff" - just much later? is the subtext that snowmelt contributes more to groundwater recharge than rainfall does?

page 7 line 22 "below until an elevation" - perhaps an extra word in here?

page 7 lines 25-26 and possibly elsewhere: "From February-May" an em dash should not stand in for the preposition "to"

page 7 lines 29-30 - again, make it clear that this is only for the MRB. "the region" should be clarified. Without further analysis outside MRB, it's mere speculation to extend these results to a wider region like the Cascades or the Northwest.

page 8 line 5 "below normal compared with historical average conditions" - could just say "below historical average conditions" unless normal means something other than historical averages, in which case specify

Figures

Figure 2 - bar charts are a difficult way to present this kind of information, and using cumulative precipitation pegs the y-axis at large values, rendering the monthly values harder to distinguish. I suggest replacing the bar charts with something more intuitive like connected line segments with symbols, and also reporting the N-M and DJF values with monthly means instead of cumulative.

Figure 3 - add the EP for the normal. Also I'm not a big fan of the format, showing the 2014 and 2015 values as horizontal lines - it's a lot of ink to convey very little information. I suggest showing just one panel with the EP curves (+0 and +2°C, perhaps for April), and replace the 6 panels with another time series showing the 2014 and 2015 snowpack, and the EP percentiles.

Figures 3, 4, 6 - are these monthly averages or first-of-month?

[Figure]

---

## Referee Comment (RC2) · Anonymous Referee #2 · 18 Jun 2016

This paper described spatial probabilistic assessment of snowpack of WY 2014 and 2015 in Mackenzie River Basin. Here are my comments on this paper.

1. The results and analysis mainly came from the SnowModel estimates. However, the authors did not provide any detail information of model (inputs, calibration & evaluation statistics). Only authors mentioned, page 5, lines 11-12, "Model forcing data include temperature and precipitation from the SNOTEL network and additional meteorological data as described in Sproles et al. (2013)." I am not sure whether authors used an exactly same framework of Sproles et al. (2013) or not. Even though authors did, authors need to provide a concise summary of the model and model performance information. Without the information, the analysis may be lost the confidence of readers.

[Figure]

2. Authors used 20-year periods (WY 1989-2009) to calculate EP with +2C condition. But the authors also presented that, page 5, line 11, "The calibration period for our model was WY2006 through WY2012." Why did authors include the period WY 2010-2012 that did not contain the experimental periods? Also, lines page 5, lines 16-18: is not clear for the reason for selecting the calibration period.

3. The authors mentioned several times in the manuscript, "extreme low snowpacks of 2013-2014 and 2014-2015." But I am confused - Page 6, lines 25-26: "For N–M in WY 2014, precipitation was at 112% of the 30-year normal and temperatures at SNOTEL stations in the MRB were 0.9 C warmer than normal." - Page 7, lines 1-2: "we see that the April 1 basin-wide snow water storage for WY 2014 corresponds to 40% EP, meaning that WY 2014 snowpack storage is slightly above average for a +2 C model perturbation." Is WY 2014 dry year in MRB?

4. As the authors mentioned that EP is generally used to show a probability of a natural hazard event occurring annually. Thus, page 6, lines 1-2, "90% EP describes the statistical likelihood of a measurement that would be met or exceeded in 90% of the time, or a 9 in 10 chance of occurring in any year, and represents a relatively low SWE value." may lead to confusion (e.g., dry season, like WY2015, may happen 90% probability in any years.) I suggest that author may use a new term or a different way of expression to clear over the entire manuscript.

5. Only 2/10 pages used to explain methods and results. Authors may more focus on their methods and results. Also, this paper pretended to perform over Pacific Northwest area. Please clearly mention that this is a case study for MBR (3,041km2) in abstract and introduction. Also, I recommended that "2 Research Methods" need to divide into two sections, study area and methods to clear.

6. Fig 2 & page 6, line 28: "30-year" – where is coming from? Authors used 20-year model estimate. But without any description, authors used 30-year average air temperature and precipitation dataset for comparison, not 20-year model inputs or estimates.

7. The authors mentioned that page 6, lines 25-26, "For the DJF period, WY 2014 monthly precipitation was 103% of normal and SNOTEL temperatures were 0.7C warmer than normal." In Fig 2., Why WY2014 for DJF is less than 30-year normal? Please check your dataset.

8. page 7, lines 7-9 – author may need to provide a figure for SWE and precipitation of WY 2014 and 2015 to clear the description.

9. page 7, lines 13-14 – "393 km2 is essentially snow-free (Fig. 4)." Where is "393 km2" coming from?

10. page 7, lines 17-19 – Authors should provide references to support their insistence.

11. page 7, line 22-23 "This ratio does not exceed 0.20 below until an elevation of 1500 m in WY 2014, which is still markedly lower than the mean SWE:P at the McKenzie SNOTEL site (0.58, 1454 m)." - Why is so much different between your estimates and SNOTEL?

12. page 8, line 1 – "slightly warmer conditions (+1–2C)" – Author already mentioned in the manuscript that 0.9C was increased in N-M for WY2014. Why +1-2C? Please mentioned specifically not an ambiguous word.

13. I have more comments, but I stop to review the manuscript. Please keep focusing on your method and results, not the general idea.

---

## Author Comment (AC1) · 28 Sep 2016

Response to referees' comments for:

*Future Snow? A Spatial-Probabilistic Assessment of the Extraordinarily Low Snowpacks of 2014 and 2015 in the Oregon Cascades*

Eric A. Sproles, Travis R. Roth, Anne W. Nolin

The authors would like to thank the referees for their time in providing comments and suggestions for this manuscript.

Referees' comments are in normal font.

*Authors' responses are in italics.*

**Referee #1**

This paper is an interesting and detailed look, using modeling, at the snow volume in two recent years in a small (3000 km2) river basin in the western Oregon Cascade mountain range. The most interesting finding may be that there was virtually no snow below about 1300m in 2015, yet this finding is buried in the discussion. It should be highlighted in the abstract.

**big picture issues**

Selecting 2014, 2015, and a notional year that is 2 C warmer than a 30-year "normal" presents a muddled message. In some places, 2014 is presented as being exceptional (e.g. page 1 lines 23-24); but in other places, it does not seem so exceptional (e.g. page 7 line 2 and the caption of Figure 3). It is a little puzzling why the EP would be shown (Fig 3) for the +2 C world but not the 30-year normal. I don't know how to un-muddle this message.

*We apologize for any muddled messages and have worked through the text to help clarify any ambiguities. WY 2014 does compare differently. If you simply focus on long-term snowpack in previous years, it was well below normal. However, if you place it in the context of +2ºC conditions, it is slightly above average. This shift to framing the conversation to a warmer world is the premise of the paper.*

*We italicized the text at the end of the methods section to reiterate this point to the reader.*

1. The paper is fairly rife with unsubstantiated assertions. I mention several below.

*Thank you for identifying portions of the text that are unsubstantiated. We have provided some additional analysis that quantifies some of our statements and in other places changed the tone of the commentary.*

2. spatial domain. much of the introduction refers to western US or PNW, and only on page 4 do we learn that the study area is actually a small ( 3000 km2) sub-watershed of the Willamette River Basin, and it is there asserted (without evidence) that the McKenzie River Basin is 'characteristic for maritime snow in the Pacific North-west'.

*Thank you for comment. We have provided some additional analysis that quantifies some of our points and in other places changed the tone of the text. We have also shifted the comparison to the Willamette Basin and away from the PNW (Fig. 1b; page 2, lines 20 – 24).*

3. The abstract should make clear where the study was conducted, and either the attempts to relate this to the PNW should be dropped or some additional analysis should be conducted. I'm not suggesting repeating the full detailed analysis on a wider domain, but the interesting finding that there was essentially no snow below 1300m in 2015 should be easy to check with SNOTEL sites throughout the Cascades (perhaps with a latitude-elevation adjustment).

*We have added that the study area is the McKenzie River basin in the abstract and introduce it contextually in the fourth paragraph of the introduction.*

minutia:

4. word choice: snow volume is variously called snow water storage, snowpack storage, and just snowpack. I suggest picking a single term and using it consistently.

*We have incorporated this suggestion throughout the paper where appropriate. Thank you for the suggestion.*

5. page 2 lines 7-9: assertions about impacts need references.

*Thank you for the suggestion, and we have added references (page 2, lines 7-11).*

6. page 2 line 16 no definition of 'critical' fits this usage - the word is often, as here, misused in place of 'critically important' or just 'important'.

*This suggestion has been incorporated into the revised version.*

7. page 3 lines 2-3 why not? do you have evidence to back up this assertion?

*These lines have been removed from the manuscript.*

8. page 3 lines 11-13 i don't follow the argument here. these deterministic approaches can also be (and have been) used to simulte past and future. the sentence seems to be arguing that "not analogous" is a weakness, but it's not clear why that's a weakness. likewise, the last sentence in the paragraph returns to the notion of limitations but it's not clear why. and in lines 22-23 again, it's not clear whether this assertion is also a criticism of the deterministic approach (i.e. "Only an analog approach...) I suggest retooling the paragraphs on lines 9-29 (and possibly the next paragraph too) to better set up the strengths and weaknesses of deterministic modeling and analog approaches. Or just drop entirely.

*We apologize for any confusion and have retooled this section of the text. (page 3, line 9 through page 4, line 18).*

9. page 3 line 34 - again, not clear whether this paragraph is describing strengths and weaknesses. the first two sentences are about analogs, but are the models referenced on line 34 the standard deterministic distributed models? Given the last 1+ page of the introduction, I was surprised to find that the paper uses a physically based snow hydrology model instead of analogs.

*We apologize for any confusion and have retooled this section of the text in a (hopefully) improved format.*

10.   page 4 lines 13-14: another unsubstantiated assertion.

*Thank you for comment. We have provided some additional analysis that quantifies some of our points and in other places changed the tone of the text. We have also shifted the comparison to the Willamette Basin and away from the PNW (Fig. 1b; page 2, lines 20 – 24).*

11.   Page 4 line 28 "most SWE" clarify "per unit area" if that is what's meant

*This suggestion has been incorporated into the revised version.*

12.   page 4 lines 30-32 these are really valuable and interesting comparisons. Need a source for the reservoir storage statement.

*Completed.*

13.   page 5 line 20-22: just call it a sensitivity test. Mote & Salathé is dated (CMIP3 vs CMIP5; see e.g. Dalton et al., Island Press 2013) and the link to IPCC is dubious, since the number discussed in the 2013 IPCC report was not exactly a "threshold set" and moreover is a global number not regional.

*This paragraph has been restructured, and the reference to Mote and Salathé has been removed (page 6, line 4 -8).*

14.   page 6 lines 3-10 no rationale is given for this re-dimensionalizing. Perhaps if the meaning of "spatial exceedance probability" was clear.

*We have retooled this paragraph (page 6, line 27 to page 7, line 2).*

15.   page 6 lines 4-5 number agreement "dimensions...has" - maybe just delete "the dimen-sions of"

*Completed.*

16. page 7 line 11 - "is greatest" for the +2 C case. Figs 4 and 5 - these are a crucial point of the paper and perhaps its main contribution. can you comment on the strengths and weaknesses of the observing network in this elevation band? is it adequate? is there evidence that these findings apply outside MRB?

*Thank you for this thoughtful insight. We have added some additional text in the introduction and discussion, as well as adding the elevation bands of the SNOTEL monitoring network in figure 4 and 6.*

*We work closely with the National Resource Conservation Service and they do a great job keeping their stations up and running. However their network was designed for different goals and normal than we have today. One of the goals of this type of paper is to push for a retooling of the monitoring network.*

17. page 7 line 19 - what does "subsequent runoff" mean? wouldn't snowfall also produce "subsequent runoff" - just much later? is the subtext that snowmelt contributes more to groundwater recharge than rainfall does?

*Thank you for your comment. We have revised the text and added a bit more context (page 8, lines 16-21 and page 10, lines 8 - 11).*

18. page 7 line 22 "below until an elevation" - perhaps an extra word in here?

*Completed.*

19. page 7 lines 25-26 and possibly elsewhere: "From February-May" an em dash should not stand in for the preposition "to"

*We have incorporated this throughout.*

20. page 7 lines 29-30 - again, make it clear that this is only for the MRB. "the region" should be clarified. Without further analysis outside MRB, it's mere speculation to extend these results to a wider region like the Cascades or the Northwest.

*Thank you for your suggestions. We have added more content and changed the way we reference things geographically in the Discussion.*

21.   page 8 line 5 "below normal compared with historical average conditions" - could just say "below historical average conditions" unless normal means something other than historical averages, in which case specify

*We have modified the text.*

Figures

22.   Figure 2 - bar charts are a difficult way to present this kind of information, and using cumulative precipitation pegs the y-axis at large values, rendering the monthly values harder to distinguish. I suggest replacing the bar charts with something more intuitive like connected line segments with symbols, and also reporting the N-M and DJF values with monthly means instead of cumulative.

*Thank you for the feedback on the figures. We have modified the upper figure (now Fig 2a) to put the cumulative values on a separate y-axis. Because we shifted the axis, we left the cumulative values instead of monthly means. We left precipitation as a bar chart, as this format is more common with precipitation.*

*For temperature (now Fig. 2b) we shifted from bar charts to connected line segments, and we think it looks much better. Thank you for the input.*

*We have also added a third sub-figure (Fig. 2c), which displays snow water volume.*

23.   Figure 3 - add the EP for the normal. Also I'm not a big fan of the format, showing the 2014 and 2015 values as horizontal lines - it's a lot of ink to convey very little informa-tion. I suggest showing just one panel with the EP curves (+0 and +2 C, perhaps for April), and replace the 6 panels with another time series showing the 2014 and 2015 snowpack, and the EP percentiles.

*Thank you for your comments. In presenting this research to water resource managers, this figure really resonates in its current format. For that reason, we have modified the horizontal lines to*

24. Figures 3, 4, 6 - are these monthly averages or first-of-month?

*We have added text to the figures as well as additional text in the captions. Thanks for pointing this out.*

**Referee #2**

This paper described spatial probabilistic assessment of snowpack of WY 2014 and 2015 in Mckenzie River Basin. Here are my comments on this paper.

1. The results and analysis mainly came from the SnowModel estimates. However, the authors did not provide any detail information of model (inputs, calibration & evaluation statistics). Only authors mentioned, page 5, lines 11-12, "Model forcing data include temperature and precipitation from the SNOTEL network and additional meteorological data as described in Sproles et al. (2013)." I am not sure whether authors used an ex-actly same framework of Sproles et al. (2013) or not. Even though authors did, authors need to provide a concise summary of the model and model performance information. Without the information, the analysis may be lost the confidence of readers.

*Thank you for your comments regarding our paper. We have added more information on the original model regarding the model framework, methods used in the calibration and validation process, and metrics regarding performance (page 5 line 27 to page 6 line 8).*

2. Authors used 20-year periods (WY 1989-2009) to calculate EP with +2C condition. But the authors also presented that, page 5, line 11, "The calibration period for our model was WY2006 through WY2012." Why did authors include the period WY 2010-2012 that did not contain the experimental periods? Also, lines page 5, lines 16-18: is not clear for the reason for selecting the calibration period.

*We have extended the analysis through 2012. Thank you for you the suggestion.*

3. The authors mentioned several times in the manuscript, "extreme low snowpacks of 2013-2014 and 2014-2015." But I am confused - Page 6, lines 25-26: "For N–M in WY 2014, precipitation was at 112% of the 30-year normal and temperatures at SNOTEL stations in the MRB were 0.9 C warmer than normal." - Page 7, lines 1-2: "we see that the April 1 basin-wide snow water storage for WY 2014 corresponds to 40% EP, meaning that WY 2014 snowpack storage is slightly above average for a +2 C model perturbation." Is WY 2014 dry year in MRB?

*We apologize for any ambiguity. The snowpacks of WY 2014 and WY 2015 were well below historical conditions, not because of precipitation but because of warmer temperatures. Temperature drives snowpack evolution and ablation in this region. We have added additional commentary regarding the warmer temperatures throughout the revised manuscript.*

*Additionally we have highlighted the text before the results that 2015 and 2015 are unperturbed, but they will be compared to +2ºC conditions.*

4. As the authors mentioned that EP is generally used to show a probability of a natural hazard event occurring annually. Thus, page 6, lines 1-2, "90% EP describes the statistical likelihood of a measurement that would be met or exceeded in 90% of the time, or a 9 in 10 chance of occurring in any year, and represents a relatively low SWE value." may lead to confusion (e.g., dry season, like WY2015, may happen 90% probability in any years.) I suggest that author may use a new term or a different way of expression to clear over the entire manuscript.

*We have modified the text to include **met or exceeded** for clarity, as this is the definition of exceedance probability.*

5. Only 2/10 pages used to explain methods and results. Authors may more focus on their methods and results. Also, this paper pretended to perform over Pacific Northwest area. Please clearly mention that this is a case study for MBR (3,041km2) in abstract and introduction. Also, I recommended that "2 Research Methods" need to divide into two sections, study area and methods to clear.

*Thank you for your comments. We have modified the text to introduce the McKenzie River basin in the abstract much earlier in the introduction. Additionally we shifted the text form the PNW to the greater Willamette River watershed. We also added a basic analysis of the hypsometry of the McKenzie and the Willamette to support our statements of similarity.*

*The study area paragraph is only one paragraph, and for this reason we kept it in the Methods section.*

6. Fig 2 & page 6, line 28: "30-year" – where is coming from? Authors used 20-year model estimate. But without any description, authors used 30-year average air temper-ature and precipitation dataset for comparison, not 20-year model inputs or estimates.

*Thank you for pointing out any ambiguity. We have added detail in the text (page 7, lines 15 –30) and in the figure captions.*

7. The authors mentioned that page 6, lines 25-26, "For the DJF period, WY 2014 monthly precipitation was 103% of normal and SNOTEL temperatures were 0.7C warmer than normal." In Fig 2., Why WY2014 for DJF is less than 30-year normal? Please check your dataset

*Thank you for pointing out this discrepancy, it has been rectified.*

8. page 7, lines 7-9 – author may need to provide a figure for SWE and precipitation of WY 2014 and 2015 to clear the description.

*We have added a third sub-figure (Fig. 2c), which displays snow water storage for the WY 2014 and 2015. We have also added this figure to the reference to the figure found in the text.*

9. page 7, lines 13-14 – "393 km2 is essentially snow-free (Fig. 4)." Where is "393 km2" coming from?

*We have added clarification (page 8, lines 12-13).*

10.    page 7, lines 17-19 – Authors should provide references to support their insistence.

*Thank you for the comment. We have modified the text in the manuscript and page 10, lines 8 - 11), however the Jefferson et al., 2008, and Tague and Grant, 2008 stand as the primary references.*

11.    page 7, line 22-23 "This ratio does not exceed 0.20 below until an elevation of 1500 m in WY 2014, which is still markedly lower than the mean SWE:P at the McKenzie SNOTEL site (0.58, 1454 m)." - Why is so much different between your estimates and SNOTEL?

*We have modified the text to improve readability (page 8, lines 23 -30).*

12.page 8, line 1 – "slightly warmer conditions (+1–2C)" – Author already mentioned in the manuscript that 0.9C was increased in N-M for WY2014. Why +1-2C? Please mentioned specifically not an ambiguous word.

*We have modified the text to only reference 2014 to +1ºC conditions (page 9, line 10).*

13.    I have more comments, but I stop to review the manuscript. Please keep focusing on your method and results, not the general idea.

*We would like to thank this referee for their time, expertise and insights. The goal of the paper was to compare the low extraordinarily low snowpacks of 2014 and 2014 to conditions that are 2ºC warmer. These goals are specifically highlighted in our introduction:*

> *- How does snow water storage from WY 2014 and WY 2015 compare to snow water storage under +2ºC conditions?*
> *- What is the probability that similar snowpacks and snow water storage will occur in the future?*
> *- How does snow water storage during WY 2014 and WY 2015 vary by elevation?*

*In our methods we describe how we accomplished these analyses. The spatial probability approach is simple and straightforward. This is reflected in the describing the methodology.*

*Snow analogs are more of a concept than a methodology, thus it is in the introduction, discussion, and conclusion.*

*Admittedly, we focus more on Discussion than model Methodology. The goal of the paper is not to re-introduce published models and papers, but rather provide new ways of applying deterministic model results to help better prepare for current climate trends. For that reason our manuscript is focused on the context and the discussion. However we do not dismiss the importance of methodology and results.*

*While we applied model results, this manuscript was not focused on the model itself. Following your suggestions we added additional descriptions to the manuscript regarding the modeling framework, calibration and validation procedure, and performance metrics.*

*The Methods and Results account for 2157 of the 5739 words (37%) in the revised manuscript. The first six paragraphs of the Discussion also o provide context to the results. Additionally the figures present our results in data rich graphics.*

*We understand your concern that a manuscript contains the proper emphasis on methods and results. And we have devoted over 1/3 of the manuscript to meet this goal. Additionally we have addressed your specific comments, and also added content to the manuscript itself.*

[revised manuscript text omitted]

E Sproles 9/22/16 1:29 PM

E Sproles 9/22/16 1:29 PM

E Sproles 9/22/16 1:29 PM

E Sproles 9/22/16 1:29 PM

[Figure]

McKenzie River Basin - Precipitation, Air Temperature, and Snow

**Figure 2:** The total precipitation (a) and mean temperatures (b) for the McKenzie River basin for water years 2014 and 2015 as compared to the 30-year normal (from the PRISM datasets). The lower figure (c) represents Basin-wide Snow Water Storage for the McKenzie River Basin for water years 2014 and 2015 and the normals (+2ºC) calculated from the 23 years used in this study. The calculations for snowpack are 5-day averages centered on the first day of each month.

E Sproles 9/22/16 1:32 PM

E Sproles 9/22/16 1:32 PM

[revised manuscript text omitted]

The warm, maritime snowpack of the Oregon Cascades is particularly sensitive to increased temperatures and approximately 51% of "at risk" snow in the PNW is in the Oregon Cascades (Nolin and Daly, 2016). As such, these two winters' extraordinarily low snowpacks offer an analog perspective for projected future snow conditions in the MRB and potentially the Willamette River Basin, with 2014 serving as an analog for slightly warmer conditions (+1ºC) and 2015 as an analog for winter temperatures increasing beyond 3ºC. April 1 snow water storage for 2014 was 470% greater than on the same date 2015. The volumetric difference between the two years (0.56 km$^3$) is 1.4 times more than the total reservoir storage capacity of the MRB (United States Army Corps of Engineers, 2016; United States Department of Agriculture, 2016).

Using spatial exceedance probability we calculate that WY 2014 maximum snow water storage was slightly above average for +2ºC conditions with an EP between 42 and 46%. By comparison, maximum snow water storage for +2ºC conditions during WY 2015 had an EP of about 92% and would be considered extraordinarily low snow years even for a +2ºC future climate scenario.

These low snow years persisted even under normal and slightly below normal annual precipitation. For N-M, precipitation was 102% (WY2014) and 81% (WY2015) of the 30-year climate normal in the MRB. 
[revised manuscript text omitted]

---

## Author Response (AR2)

Response to referees' comments (November 5, 2016) for:

*Future Snow? A Spatial-Probabilistic Assessment of the Extraordinarily Low Snowpacks of 2014 and 2015 in the Oregon Cascades*

Eric A. Sproles, Travis R. Roth, Anne W. Nolin

The authors would like to thank the referee for her/his time in providing comments and suggestions for this manuscript.

Referees' comments are in normal font.

*Authors' responses are in italics.*

Comments to the Author:
Dear authors, the key reviewer for this paper and I are satisfied that the revised m/s is suitable for publication in TC subject to addressing the minor issues raised below. I sent you a copy of the GRL paper cited by the reviewer.

**Reviewer comments:**

I find the manuscript satisfactorily improved. In particular, the text is more compelling and better-referenced; figures 2 and 4 are better, and the addition of Figure 1b is helpful.

*Thank you for your feedback.*

A couple of remaining minor issues:

Figure 3 has changed, and many of the exceedance curves are less smooth than before - the caption mentions that this is now a 5-day average, but I don't see either what prompted the change (in reviewer comments - maybe it was from an online commenter not directly addressed) or what methodological difference might explain it.

*The data used for the calculations has always been the 5-day mean. This approach was applied in the original methods, so nothing has changed with regards to the calculations. We added text to the figure captions, so the 5-day mean is not buried in the methods.*

*The same data were used in the figures. The additional water years (2010 – 2012) caused the "less smooth" curves.*

There's a paper just published that might be worth including, on the 2015 snowpack in the western US: GRL doi:10.1002/2016GL069965

*Thank you for including this paper, we have added it to the text.*

The text needs another careful editorial pass; I found several remaining (and new) examples of sentences that didn't quite parse, whether from words that are extra or missing or out of order. Reading aloud is sometimes helpful at finding those sentences.

*Thank you for your comments and feedback.*

[revised manuscript text omitted]

[Figure]

Figure 2: The total precipitation (a) and mean temperatures (b) for the McKenzie River Basin for water years 2014 and 2015, as compared to the 30-year normal (from the PRISM datasets). The lower figure (c) represents Basin-wide Snow Water Storage for the McKenzie River Basin for water years 2014 and 2015 and the normals (+2ºC) calculated from the 23 years used in this study. The calculations for snowpack are 5-day averages centered on the first day of each month.

Sproles 11/10/2016 3:48 AM

[Figure]

**Figure 3: The exceedance probability of basin-wide snow water storage under +2°C conditions. During 2014, snow water storage increased considerably in March to reach above average conditions. The snowpack during the winter of 2015 was extremely low, and never increased beyond 0.21 km³. The calculations are 5-day averages centered on the first day of each month.**

[Figure]

**Figure 4: Volumetric snow water storage binned by 50 m elevation bands. The corresponding basin-wide snow water storage (km³) for 2014 and 2015 is provided for each month. Larger snowpacks (lower exceedance probability) have considerable contributions between 1000 and 1300 m. During 2014 and 2015, this elevation range had minimal snowpack, despite close to normal precipitation. Note that on the vertical axes, snow water storage below 500 m and above 2500 m are not included for visual clarity. These elevations contribute minimally to basin-wide snow water storage. The calculations are 5-day averages centered on the first day of each month.**

Sproles 11/10/2016 3:48 AM

Sproles 11/10/2016 3:48 AM

[Figure]

**Figure 5: The spatial distribution of SWE on April 1 from water years 2014 and 2015 as compared to the corresponding EP. Both the distribution and magnitude of SWE are strikingly similar. The calculations are 5-day averages centered on the first day of each month.**

Sproles 11/10/2016 3:48 AM

[Figure]

**Figure 6: The ratio of SWE:P binned by 50 m elevation bands. The relationship between elevation and SWE:P is evident across all exceedance probabilities. Under +2°C simulations and in 2014 and 2015, roughly 1500 m is the elevation at which SWE:P begins to increase substantially along the horizontal axis. Note that on the vertical axes, snow water storage below 500 m and above 2500 m are not included for visual clarity. These elevations contribute minimally to basin-wide snow water storage. The calculations are 5-day averages centered on the first day of each month.**